# Mitigating Distribution Shifts: Uncertainty-Aware Offline-to-Online Reinforcement Learning

## Abstract

Deploying reinforcement learning (RL) policies in real-world scenarios, particularly through offline learning approaches, faces challenges due to distribution shifts from training environments. Past approaches have shown limitations such as poor generalization to out-of-distribution (OOD) variations or requiring extensive retraining on target domains. We propose Uncertainty-aware Adaptive RL, UARL, a novel offline RL pipeline that enhances OOD detection and policy generalization without directly training in OOD environments. UARL frames distribution shifts as OOD problems and incorporates a new OOD detection method to quantify uncertainty. This approach enables iterative policy fine-tuning, starting with offline training on a limited state space and progressively expanding to more diverse variations of the training environment through online interactions. We demonstrate the effectiveness and robustness of UARL through extensive experiments on continuous control tasks, showing reliability in OOD detection compared to existing method as well as improved performance and sample efficiency.

## 1 Introduction

Robust Reinforcement Learning (RL) offers optimal solutions under relatively idealized theoretical conditions (Iyengar, 2005; Nilim & El Ghaoui, 2005; Tamar et al., 2014). However, deploying these RL policies in real-world applications poses substantial safety concerns, as real-world environments often deviate from theoretical assumptions (Dulac-Arnold et al., 2021; García & Fernández, 2015). The adoption of RL in real-world applications, such as robotics (Kober et al., 2013) and industrial control (Spielberg et al., 2019), has highlighted the importance of addressing these challenges.

The online nature of RL, requiring continuous interaction with the environment, often proves impractical in real-world settings due to associated risks and costs (Sutton & Barto, 2018; Levine et al., 2020). This challenge is compounded by Out-Of-Distribution (OOD) events, such as sensor noise or unmodeled environmental changes, which can substantially degrade the performance of trained policies (Zhao et al., 2020; Huang et al., 2023; Danesh & Fern, 2021). Existing research on robustness and safety provides a foundation for tackling these issues, including robust control strategies that optimize policies to handle worst-case scenarios (Iyengar, 2005; Nilim & El Ghaoui, 2005). However, it introduces new challenges, particularly the distributional shift between the data collection policy and the learned policy, which can be exacerbated during online fine-tuning (Lee et al., 2022; Zheng et al., 2023; Zhang et al., 2023a).

Domain randomization (DR) techniques enhance robustness by training in simulation and deploying the policy to real-world environments (Tobin et al., 2017). This approach has emerged as an alternative, allowing training on pre-collected static datasets (Levine et al., 2020) from perturbed environments in simulation. However, determining how to design the simulated environment to reflect real-world variability accurately is often challenging. In addition, domain randomization introduces significant safety concerns when testing potentially unsafe policies on physical hardware (Mehta et al., 2020). Despite these advances, developing policies that ensure safety under unexpected real-world conditions remains relatively underexplored in RL. Accelerating research on methods addressing these challenges is crucial for enabling safe real-world deployment of RL systems.

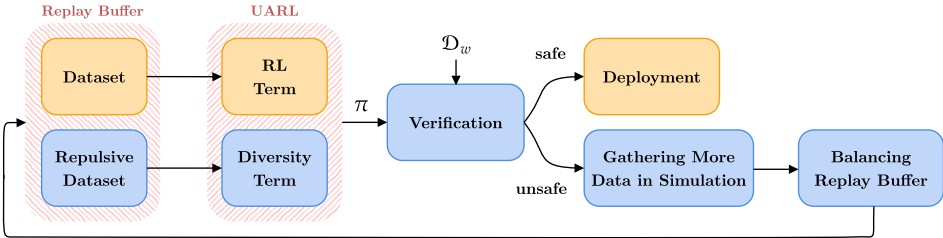

Figure 1: Overview of UARL framework. Blue boxes show our contributions while orange boxes are adapted from existing RL methods. Agent processes nominal and repulsive datasets through RL and diversity terms. The verification module assesses policy safety, guiding deployment, or additional data collection. This process helps to manage OOD scenarios.

In this paper, we introduce **U**ncertainty-aware **A**daptive **RL** (UARL), a novel approach that encourages safe policy deployment under distribution shift without direct access to the target environment. As shown in Fig. 1, UARL introduces a diversity term that consists of an ensemble of critics to quantify policy uncertainty and iteratively refines high-uncertainty regions (of the state space). UARL integrates nominal and repulsive datasets in the replay buffer and employs a verification module for safe adaptation to OOD scenarios. We validate UARL on MuJoCo benchmark tasks (Todorov et al., 2012), assessing performance, safety, and sample efficiency.

**Contributions**. Our **key contributions** are (1) a method for **quantifying uncertainty and adapting policy without direct interactions in the OOD environments**, and (2) an efficient offline-to-(*semi*)online (O2O) RL strategy to **balance the replay buffer**. Unlike traditional O2O methods that transition directly from offline learning to online fine-tuning in the target environment, our approach leverages simulated environments to gradually expose the policy to increasing levels of environmental variability, eliminating the need to fine-tune policies with immediate real-world interaction. Compared to domain randomization, our method avoids the risk of validating potentially unsuitable policies in OOD environments, thereby improving the safety and reliability of RL in real-world applications. These contributions collectively advance the field of safe and robust RL, offering a novel framework that enhances policy generalization and safety across diverse simulated environments, while providing a foundation for addressing the challenges of distribution shifts and OOD events in RL[1].

## 2 RELATED WORK

In this section, we provide a summary of related work in offline RL, offline-to-online RL, and ensemble methods. A comprehensive discussion of the Related Work is presented in App. C.

**Offline RL**. Traditional RL allows policies to interact freely with environments to discover optimal strategies (Sutton & Barto, 2018). In contrast, offline RL addresses scenarios where online interaction is impractical or risky, learning solely from pre-collected datasets gathered by a behavior policy (Fujimoto et al., 2019; Agarwal et al., 2020; Ernst et al., 2005; Kumar et al., 2019; Levine et al., 2020; Lange et al., 2012; Kostrikov et al., 2022; Wang et al., 2020; Tarasov et al., 2024). This approach enables applications in domains where real-time learning might be unsafe. However, offline RL faces a significant challenge in *distributional shift*, where the training data may not accurately represent the deployment environment, potentially leading to suboptimal or unsafe behavior. Several approaches address this issue by encouraging the learned policy to resemble the behavior policy (Jaques et al., 2019; Wu et al., 2019; Siegel et al., 2020; Fujimoto & Gu, 2021), promoting caution through action alignment. Other methods explore training conservative critics for more cautious reward estimates (Kumar et al., 2020; Kostrikov et al., 2021) or diversifying critics within the actor-critic framework to improve robustness (An et al., 2021; Bai et al., 2022; Wu et al., 2021).

**Offline-to-Online RL**. Building upon offline RL, O2O RL leverages previously collected offline datasets to accelerate online RL training. This approach first pre-trains a policy with offline RL and then continues to fine-tune it with additional online interactions (Nair et al., 2020; LEI et al., 2024; Zhao et al., 2024; Zheng et al., 2022). However, naive O2O RL is often unstable due to the

---

[1]Anonymized source code is available at: anonymous.4open.science/r/UARL-41CD

distributional shift between the offline dataset and online interactions. To address this, researchers have proposed techniques such as balanced sampling (Lee et al., 2022), actor-critic alignment (Yu & Zhang, 2023), adaptive conservatism (Wang et al., 2024), return lower-bounding (Nakamoto et al., 2023), adaptive update strategies (Zheng et al., 2023), introducing online policies alongside offline ones (Zhang et al., 2023a), and using weighted replay buffers with density ratio estimation (Lee et al., 2022). While these works primarily focus on maximizing cumulative rewards and addressing distributional shifts, our paper explicitly considers policy safety, particularly for OOD samples. Our work builds upon weighted samples but offers a more efficient solution by leveraging learned critics to assign weights to offline and online samples during fine-tuning.

**Ensemble Methods** combine multiple models to enhance performance and are widely used in ML applications (Goodfellow et al., 2016). In OOD detection, ensembles offer robustness by leveraging model diversity (Lee & Chung, 2020; Lakshminarayanan et al., 2017). While random initialization contributes to diversity, controlled diversity is often achieved by manipulating the loss function (Wabartha et al., 2020; Mehrtens et al., 2022; Jain et al., 2020; Pang et al., 2019). This allows for fine-tuning ensemble behavior, particularly through regularization techniques or bias-variance decomposition (Wood et al., 2023; Arpit et al., 2022), enhancing both ID accuracy and OOD detection by ensuring disagreement in uncertain regions, known as "repulsive locations" (Hafner et al., 2020). In RL, ensembles play a key role in optimizing exploration strategies. Osband et al. (2016) introduced ensemble critics for more efficient exploration, and Lee et al. (2021) proposed leveraging Q-ensembles to augment Q-value estimates using the mean and standard deviation of the ensemble. These methods also support uncertainty estimation, crucial for handling OOD scenarios in RL, by analyzing discordance between ensemble members' predictions (Wabartha et al., 2020; Liu et al., 2019; Lakshminarayanan et al., 2017; Jain et al., 2020). Building on these foundations, we employ critic disparity in actor-critic algorithms to effectively detect OOD instances, combining the strengths of ensemble methods in both OOD handling and RL optimization.

## 3 BACKGROUND

### 3.1 REINFORCEMENT LEARNING

RL problems commonly model the world as a Markov Decision Process (MDP) $M = \langle S, A, T, R, \gamma \rangle$, where $S$ is state space, $A$ is action space, $T(s'|s,a)$ is the state transition function, $R(s,a)$ is the reward function and $\gamma \in [0,1)$ is discount factor (Bellman, 1957; Sutton & Barto, 2018). In RL, the objective is to find the optimal policy that maximizes the expected cumulative return: $\pi^* = \arg\max_\pi \mathbb{E}_{s;a\sim\pi(\cdot|s)} \left[ \sum_{t=0}^\infty \gamma^t R(s,a) \right]$ with $\pi(a|s)$ representing the probability of taking $a \in A$ in $s \in S$ under policy $\pi$. While online RL algorithms can be either on-policy (updated based on data from the current policy) or off-policy (updated based on data from any policy), offline RL is inherently off-policy. In offline RL, the policy is learned using a static, pre-collected dataset $\mathcal{D} = \{(s,a,s',r)\}$ obtained by a behavior policy $\pi_b$, where $r$ is the immediate reward obtained after taking $a \in A$ in $s \in S$ and transitioning to $s' \in S$ (Levine et al., 2020). Given dataset $\mathcal{D}$, the Q-function update rule during policy iteration is defined as:

$$Q_{k+1}^\pi \leftarrow \arg\min_Q \mathbb{E}_{s,a,r,s'\sim\mathcal{D}} \left[ (Q(s,a) - (r + \gamma\mathbb{E}_{a'\sim\pi_k(\cdot|s')}[Q_k^\pi(s',a')]))^2 \right] \quad \text{policy evaluation} \quad (1)$$

which updates the Q-values by minimizing the MSE between the current Q-values and the target values, approximating the true Q-values that satisfy the Bellman equation under the current policy. The policy $\pi$ is then updated towards actions that maximize the expected Q-value:

$$\pi_{k+1}(\cdot|s) \leftarrow \arg\max_\pi \mathbb{E}_{s\sim\mathcal{D},a\sim\pi_k(\cdot|s)}[Q_{k+1}(s,a)] \quad \text{policy improvement} \quad (2)$$

By iteratively evaluating and improving the policy, with appropriate assumptions, actor-critic RL converges to a near-optimal policy maximizing the expected cumulative return (Levine et al., 2020). However, a key challenge in offline RL is the distributional shift between the dataset $\mathcal{D}$ and the state-action distribution induced by the learned policy, potentially leading to overestimation of Q-values for state-action pairs not well-represented in the dataset $\mathcal{D}$, potentially resulting in poor performance when the learned policy is deployed (Kumar et al., 2020).

To mitigate this, approaches like conservative learning employ critics that lower-bound the true value (Kumar et al., 2020; Fujimoto et al., 2019; Kumar et al., 2019). Another approach is ensemble

diversification, which involves increasing the number of Q-networks and diversifying them (An et al., 2021). This diversification is achieved by minimizing the cosine similarity between the gradients of different critics with respect to their inputs, encouraging the critics to capture different aspects of the value function. These methods aim to mitigate distribution shifts and reduce uncertainty in offline RL by modifying the policy evaluation step. The challenges of distributional shift are further exacerbated when fine-tuning offline RL policies in an online setting (O2O RL) (Zheng et al., 2023; Lee et al., 2022; Zhang et al., 2023a). In this scenario, the policy encounters previously unseen states during online interaction, potentially leading to inaccurate Q-value estimates. This can harm the performance of the initial policy learned offline, especially when the offline data has limited coverage of the state-action space.

## 3.2 Ensemble Diversification

As detailed in Sec. 2, ensembles show improved efficacy when supplemented with an extra loss term, beyond initial weight randomization, to regulate diversity. The DENN method (Wabartha et al., 2020) implements this approach by introducing a new diversity term, $\mathcal{L}_{\text{div}}$, into the loss function. This method utilizes the concept of **repulsive locations**, strategically selected data points designed to induce disagreement among ensemble models, particularly at the boundary of the training distribution, where the model's predictions are less certain and may exhibit higher variance.

Let $\mathcal{X} = \{x_1, x_2, ..., x_n\}$ denote the inputs and $Y$ be the corresponding output space of the nominal dataset. One approach to defining repulsive locations is to add noise to the input data: $\mathcal{X}' = \{x + \epsilon : x \in \mathcal{X}\}$, where $\epsilon$ represents noise. Alternatively, an entirely different dataset can serve as repulsive locations (Hendrycks et al., 2019). By introducing these repulsive locations, uncertainty is enforced at the boundary of the training distribution, effectively propagating into OOD regions, thereby enhancing OOD detection crucial for robust model performance. To leverage these repulsive locations, DENN employs an ensemble of multiple models, each denoted as $f_i : \mathcal{X} \rightarrow Y$. DENN promotes diversity by constraining each $f_i$ to differ from a reference function $g : \mathcal{X} \rightarrow Y$, which was trained once on the nominal dataset. This reference function $g$ serves as a consistent baseline for diversity promotion. DENN augments the conventional supervised learning loss function by:

$$\mathcal{L}(f_i, g, \mathcal{X}, \mathcal{X}') = \frac{1}{|\mathcal{X}|} \sum_{x,y \in \mathcal{X}} (f_i(x) - y)^2 + \underbrace{\frac{\lambda}{|\mathcal{X}'|} \sum_{x \in \mathcal{X}'} \exp(-||f_i(x) - g(x)||^2/2\delta^2)}_{\text{diversity term } \mathcal{L}_{\text{div}}} \quad (3)$$

where $\lambda$ is the diversity coefficient and $\delta$ controls the diversity between two models at data point $x \in \mathcal{X}'$. The diversity term $\mathcal{L}_{\text{div}}$ penalizes the similarity between $f_i$ and the reference function $g$, which leads to different predictions at inputs $\mathcal{X}'$, thus making $f_i$ diverse with respect to $g$.

## 4 Uncertainty-aware Adaptive RL

In this section, we start by defining our repulsive locations in Sec. 4.1. We then outline our key technical elements, including an ensemble of critics for OOD detection (Sec. 4.2), a balanced replay buffer to handle distribution shifts (Sec. 4.3), and an iterative policy refinement process (Sec. 4.4).

### 4.1 Repulsive Locations in RL

While repulsive locations enhance model diversity in supervised learning (Wabartha et al., 2020), applying them in sequential decision-making requires a more nuanced approach than simply adding noise to data. To address the challenge of real-world distribution shifts, we generate repulsive locations by injecting variability into environment dynamics through hyperparameter randomization. We select key hyperparameters like initial noise scale, friction coefficient, and agent's mass, inspired by common variations in real-world tasks that we have no accurate knowledge about.

We assume that we do not have direct knowledge of the target environment $E_w$. We begin with an initial environment $E_0$ (which may not align with $E_w$) and a repulsive location $E_1$ through parameter randomization. By progressively expanding the randomization range, we create a curriculum that

gradually introduces more randomized scenarios. This approach aims to push the agent's exploration towards the real-world environment, $E_w$.

Fig. 2 is a conceptual illustration on how UARL progressively expands the randomization range from $E_0$ to $E_n$, where $E_i$ for $i \in \mathbb{N}$ and $0 < i \leq n$ denotes the environment with increasingly diverse environmental conditions. Importantly, $E_{w'}$ represents a subset of the $E_w$ that is sufficient for the agent to perform well in the target environment. Our goal is to train the agent to operate effectively within $E_{w'}$, balancing comprehensive coverage of likely scenarios with computational efficiency.

By focusing on this practical subset rather than the entire state space, or a wide blindly chosen one like in domain randomization (Tobin et al., 2017), we aim to achieve safe policy deployment without the computational cost of exhaustive exploration. Sec. 5 provides details on these randomization procedures and how they contribute to the agent's adaptability.

Figure 2: A conceptual visualization of state space expanding from $E_0$ (initial) to $E_1$, $E_2$, and $E_3$ by increasing randomization. DR denotes domain randomization, $E_w$ is the theoretical state space, and $E_{w'}$ is the region in which the agent can perform effectively.

### 4.2 OFFLINE RL WITH DIVERSE CRITICS

We extend DENN's diversity term $\mathcal{L}_{\text{div}}$ to not only have diverse critics but also be able to use critics to estimate uncertainty in the environment (Wabartha et al., 2020). This uncertainty estimation is crucial for solving the problem of safe and robust RL in real-world applications, as it allows the agent to identify and adapt to situations where its knowledge is limited or potentially unreliable.

When extending DENN to RL, we need to consider that, unlike supervised learning, data in RL are not labelled. Therefore, we need to define a reference function that corresponds to the value each critic in the ensemble would naturally predict OOD. An interesting observation is that we do not need to pretrain the reference function since we have the Bellman target value as the label. In our approach, we learn an ensemble of $N$ distinct Q-functions, denoted as $\{Q_1, Q_2, \ldots, Q_N\}$. With that, we modify $\mathcal{L}_{\text{div}}$ in Eq. 3 to the following:

$$\mathcal{L}_{\text{div}}^{\text{RL}} = \sum_i \exp\left(-||Q_i(s,a) - (r + \gamma Q_i(s', \pi(a'|s')))||^2 / 2\delta^2\right); (s, a, s', r) \sim \mathcal{D}' \tag{4}$$

where in the offline RL setting, $\mathcal{D}'$ is the repulsive dataset which can be defined as the dataset gathered by the behavior policy over any modified version of the original environment, which we detail in Subsec. 4.1. Based on temporal difference learning, we can consider $r + \gamma Q_i(s', \pi(a'|s'))$ as the learner's target value, and compare that against the predicted value of $Q_i(s,a)$ (Sutton & Barto, 2018). By combining Eq. 1 and Eq. 4, our overall policy evaluation step for each $Q_i$ in the ensemble:

$$Q_{i,k+1}^{\pi} \leftarrow \underset{Q_i}{\arg\min} \, \mathbb{E}_{s,a,s',r \sim \mathcal{D}} \left[ (Q_i(s,a) - (r + \gamma Q_i^{\pi}(s', \pi(a'|s'))))^2 \right]$$

$$+ \lambda(\underbrace{\mathbb{E}_{s,a,s',r \sim \mathcal{D}'; a' \sim \pi(\cdot|s')} \left[ \sum_i \exp(-||Q_i(s,a) - (r + \gamma Q_i(s', a'))||^2 / 2\delta^2) \right]}_{\text{diversity term } \mathcal{L}_{\text{div}}^{\text{RL}}}) \tag{5}$$

Eq. 4 and Eq. 5 form the core of our work, promoting diversity among the Q-functions in our ensemble. The diversity term $\mathcal{L}_{\text{div}}^{\text{RL}}$ in Eq. 5 encourages each $Q_i$ to diverge from its own Bellman target on the repulsive dataset $\mathcal{D}'$, while the first term ensures accurate Q-value estimation on the nominal dataset $\mathcal{D}$. This formulation is analogous to Eq. 3, where each model in the ensemble is encouraged to differ from a reference function.

The diversity term in our formulation is key to improving robustness and uncertainty estimation in our ensemble. The update rules (Eq. 4 and Eq. 5) promotes agreement amount Q-functions on $\mathcal{D}$ and diversity on $\mathcal{D}'$, providing a clearer separation between $\mathcal{D}$ (ID) and $\mathcal{D}'$ (OOD). Without Eq. 4 and Eq. 5, a set of Q-functions with minimal diversity would produce similar values across both nominal and repulsive datasets. This lack of diversity would diminish the impact of $\mathcal{L}_{\text{div}}^{\text{RL}}$, as the exponential term would stay close to 1, limiting the disagreement between $Q_i(s,a)$ and $(r + \gamma Q_i(s', \pi(a'|s')))$. As a result, the ensemble would struggle to capture a range of Q-value estimates, especially in OOD scenarios, leading to poor uncertainty estimation and overconfidence in unfamiliar situations.

---

**Algorithm 1** Balancing replay buffer

---

1: **Require:**                                                                 ▷ *Before balancing*
    ensemble of critics $Q$, online dataset $\mathcal{D}^{\text{on}}$, offline dataset $\mathcal{D}^{\text{off}}$.

2: **for** each $(s, a) \in \mathcal{D}^{\text{on}} \cup \mathcal{D}^{\text{off}}$ **do**                    ▷ *Compute variance for every data point*
3:     Compute the variance $\sigma^2(s, a)$ between the critics' outputs $Q(s, a)$
4:     **if** $s \in \mathcal{D}^{\text{on}}$ **then**
5:         Assign weight $w$ proportional to $\frac{1}{\sigma^2(s,a)}$                ▷ *Higher weight to ID samples*
6:     **else if** $s \in \mathcal{D}^{\text{off}}$ **then**
7:         Assign weight $w$ proportional to $\sigma^2(s, a)$                      ▷ *Higher weight to OOD samples*

---

The objectives pursued in the policy evaluation step exhibit conflicting tendencies if $\mathcal{D} \cap \mathcal{D}' \neq \emptyset$, resulting in regularization for small enough values of $\lambda$ (Szegedy et al., 2016). Beyond $\mathcal{D}$, $\mathcal{L}_{\text{div}}^{\text{RL}}$ becomes the dominant term, encouraging OOD repulsion between $Q_i(s, a)$ and $(r + \gamma Q_i(s', a'))$, thus promoting diverse Q-values OOD. It is noteworthy that when $\lambda = 0$, Eq. 5 returns to the standard RL loss. Each critic within our ensemble is trained to utilize Eq. 5. Given the stochastic initialization of their respective parameters, the critics will naturally develop different predictions, especially in regions of the state-action space that are not well-represented in the training data.

## 4.3 FINE-TUNING POLICY WITH BALANCING REPLAY BUFFER

To address some of the existing problems in offline RL settings, recent work has shifted towards an O2O approach, where policies are pre-trained with offline data and refined through online interactions, as discussed in Sec. 2. However, effectively utilizing both offline and online datasets requires consideration of their disparate distributions, which may hinder policy fine-tuning (Nakamoto et al., 2023). To stabilize fine-tuning and expedite convergence, we propose assigning weights to samples based on the critics' uncertainty about them. We define the weight of a sample $(s, a, r, s')$ as:

$$w(s, a, r, s') = \begin{cases} \frac{1}{\sigma^2(Q_1(s,a), ..., Q_N(s,a))} & \text{if } (s, a, r, s') \in \mathcal{D}^{\text{on}} \\ \sigma^2(Q_1(s, a), ..., Q_N(s, a)) & \text{if } (s, a, r, s') \in \mathcal{D}^{\text{off}} \end{cases} \qquad (6)$$

where $Q_1, ..., Q_N$ are the $N$ critics in our ensemble, and $\sigma^2(\cdot)$ denotes the variance. This approach motivates a balanced replay scheme that manages the trade-off between utilizing online samples, beneficial for adaptation, and offline samples, stable for maintaining baseline performance. The pseudocode for balancing the replay buffer is outlined in Alg. 1.

## 4.4 UARL ALGORITHM

Our approach assumes access to a limited real-world demonstration dataset $\mathcal{D}_w$ as a proxy for the target deployment environment. This enables us to evaluate policy uncertainty on real-world data without risking unsafe deployments. Collected from a few demonstrations, $\mathcal{D}_w$ captures key aspects of the real-world task that may differ from simulation.

Initially, we execute a behavior policy for $n$ episodes in an unaltered simulation environment $E_0$, generating an offline dataset $\mathcal{D}_0$. We then introduce slight randomization to a single hyperparameter in the environment $E_1$, collecting a repulsive dataset $\mathcal{D}_1$ using the same behavior policy. These two datasets, $\mathcal{D}_0$ (nominal) and $\mathcal{D}_1$ (repulsive), are used in our loss function: $\mathcal{D}_0$ contributes to the standard RL loss, while $\mathcal{D}_1$ informs the diversity term $\mathcal{L}_{\text{div}}^{\text{RL}}$ (Eq. 5).

After training, the variance of the critic ensemble over a given dataset serves as a critical metric in this process. As defined in Eq. 5, the loss function promotes agreement amount critics on $\mathcal{D}$ and disagreement on $\mathcal{D}'$. Consequently, lower variance indicates that the data is ID, while high variance suggests the data is OOD. For this, we evaluate the learned critics' uncertainty on $\mathcal{D}_w$. If the uncertainty is below a predefined threshold (Subsec. 5.2), then $\mathcal{D}_w$ must have aligned with $\mathcal{D}_0$, and the policy is deemed ready for deployment.

Otherwise, we expand the randomized hyperparameters from $E_1$ to $E_2$ and collect the new repulsive dataset $\mathcal{D}_2$. We combine $\mathcal{D}_0$ and $\mathcal{D}_1$ as the nominal dataset using our balancing replay buffer method (Subsec. 4.3) and apply the O2O approach to refine the policy. This process continues iteratively until

---

**Algorithm 2** UARL

---

1: **Require:** ▷ *Before training*
    real-world dataset $\mathcal{D}_w$, behavior policy $\pi_b$, original environment $E_0$, threshold.

2: $\mathcal{D}_0 \leftarrow$ rollouts of $\pi_b$ over $E_0$ ▷ *Subsec. 4.1*
3: $E_1 \leftarrow$ expanding $E_0$ by increasing the parameter randomization range
4: $\mathcal{D}_1 \leftarrow$ rollouts of $\pi_b$ over $E_1$

5: Train policy $\pi_0$ and ensemble of $N$ critics $Q_0$ with Eq. 5 and Eq. 2 with nominal and
    repulsive datasets: $(\mathcal{D}_0, \mathcal{D}_1)$ ▷ *Subsec. 4.2*
6: $\sigma^2 \leftarrow \frac{1}{N} \sum_{j=0}^{N-1} \left( Q_{0_j} - \frac{1}{N} \sum_{k=0}^{N-1} Q_{0_k} \right)^2$
7: $i \leftarrow 0$

8: **while** $\sigma^2 >$ threshold **do** ▷ *Continue until policy is safe*
9:     $E_{i+2} \leftarrow$ expanding $E_{i+1}$ by increasing the parameter randomization range ▷ *Subsec. 4.1*
10:     $\mathcal{D}_{i+2} \leftarrow$ rollouts of $\pi_i$ over $E_{i+2}$
11:     $\mathcal{D}_{nom} \leftarrow$ balanced replay buffer with $\mathcal{D}_0, \mathcal{D}_1, ..., \mathcal{D}_{i+1}$ ▷ *Subsec. 4.3 - Alg. 1*
12:     Finetune $\pi_{i+1}$ and $Q_{i+1}$ with Eq. 5 and Eq. 2 with nominal and
        repulsive datasets: $(\mathcal{D}_{nom}, \mathcal{D}_{i+2})$ ▷ *Subsec. 4.2*
13:     $\sigma^2 \leftarrow \frac{1}{N} \sum_{j=0}^{N-1} \left( Q_{i+1_j} - \frac{1}{N} \sum_{k=0}^{N-1} Q_{i+1_k} \right)^2$
14:     $i \leftarrow i + 1$
15: Deploy policy $\pi_i$
    ▷ *Detailed hyperparameter explanations found in the App. A.1*

---

the uncertainty criterion is satisfied for deployment. Alg. 2 outlines the iterative fine-tuning process, leading to a policy with sufficient certainty for deployment.

Note that if the condition in line 8 of Alg. 2 is never violated, it suggests ineffective domain randomization. High uncertainty despite varied training environments indicates that the agent has not generalized to real-world conditions, possibly due to inadequate randomization or a mismatch with real-world data. In such cases, training should be stopped, and the domain randomization or real-world data needs revisiting. This is not a shortcoming of UARL, but a limitation of the domain randomization or data quality.

Comparing other work on uncertainty-aware RL, our work explicitly identifies scenarios when the policy encounters OOD situations without direct interaction in the target environment. This is a crucial safety feature for real-world deployment. Additionally, the progressive nature also allows the agent to adapt to increasingly complex environments while maintaining performance stability.

Our iterative training pipeline works with any off-policy RL algorithm using an ensemble of critics, offering a flexible framework for improving robustness and safety in real-world RL. Our formulation presented in Eq. 5 solely introduces a "diversity term" into the policy evaluation step; therefore, UARL seamlessly integrates into any offline RL algorithm. In Sec. 5, we incorporated UARL into CQL (Kumar et al., 2020), AWAC (Nair et al., 2020), and TD3BC (Fujimoto & Gu, 2021).

## 5 EXPERIMENTS

Our evaluation comprehensively assesses the performance, robustness, and efficiency of the proposed UARL approach. We compare it against several state-of-the-art offline and O2O RL methods to address the following key questions: (1) Does UARL impact the performance of baseline methods? (Subsec. 5.1) (2) How effectively does UARL differentiate between ID and OOD samples? (Subsec. 5.2) (3) What is the effectiveness of the balancing replay buffer mechanism in UARL? (Subsec. 5.3) (4) How sample-efficient is UARL? (App. B.4)

**Baselines**. We benchmark UARL against the following prominent offline and O2O RL methods: CQL (Kumar et al., 2020), which learns conservative value estimations to address overestimation issues; AWAC (Nair et al., 2020), which enforces policy imitation with high advantage estimates; TD3BC (Fujimoto & Gu, 2021), an offline RL method that combines TD3 (Fujimoto et al., 2018) with behavioral cloning; and EDAC (An et al., 2021), which learns an ensemble of diverse critics by minimizing the cosine similarity between their gradients. We implement these baselines based on the

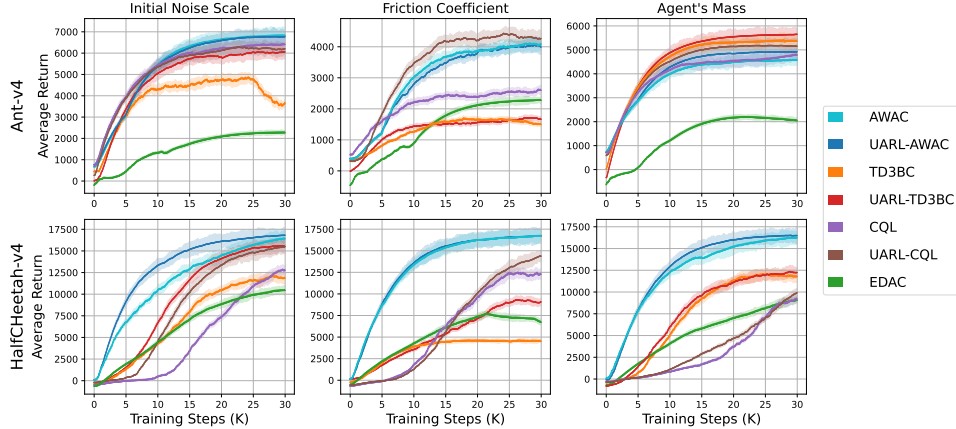

Figure 3: Offline training (1$^{\text{st}}$ iteration) performance, showing average return during training across the randomized hyperparameters. Curves are smoothed for clarity.

CORL framework (Tarasov et al., 2022) without additional hyperparameter tuning. UARL is applied to CQL, AWAC, and TD3BC, but not to EDAC due to its inherent diversity mechanism.

**Evaluation Criteria**. Our comprehensive evaluation framework for UARL focuses on performance, robustness, and sample efficiency. We introduce systematic randomization to three key hyperparameters in each environment: initial noise scale, friction coefficient, and the agent's mass. By isolating the effect of each hyperparameter, we assess the method's adaptability to dynamic conditions. Specifically, the noise scale is multiplied by $10^2$, while both the friction coefficient and the agent's mass are increased proportionately per iteration (App. A.2 provides further details). To ensure robust results, we aggregate the outcomes of 5 random seeds, reporting the mean and a 95% confidence interval. We evaluate the baselines and UARL based on the following criteria: (1) Cumulative return on evaluation environments during training, which measures overall performance; (2) OOD accuracy, defined as the critic variance's ability to differentiate ID and OOD environments; (3) Sample efficiency, defined as the number of samples required to reach a pre-defined return threshold.

**Dataset**. Although the D4RL dataset (Fu et al., 2020) is widely used in offline RL, it lacks behavior policy checkpoints needed for generating the repulsive dataset required by UARL. Thus, we created a new dataset using arbitrary behavior policies. Key differences from D4RL include: (1) We use MuJoCo v4 with Gymnasium, fixing bugs in the v2 version used by D4RL (Towers et al., 2023); (2) We extend beyond D4RL's three locomotion environments to include Ant and Swimmer; (3) For the behavior policy, we train SAC (Haarnoja et al., 2018) until convergence, generating 999 rollouts per environment. Despite these changes, our dataset's metrics closely match those of D4RL. Therefore, we expect our findings to generalize to D4RL once the necessary behavior policies become available. The real-world dataset $\mathcal{D}_w$ is obtained by running the behavior policy in the environment with a wide range of randomized parameter values, ensuring reflecting real-world variability.

**Diversity Loss Hyperparameters**. Selecting the hyperparameters $\lambda$ and $\delta$ in the diversity loss term, $\mathcal{L}_{\text{div}}^{\text{RL}}$ (Eq. 5), is key to balancing the RL objective with promoting diversity among critics. We set $\lambda$ adaptively such that the $\mathcal{L}_{\text{div}}^{\text{RL}}$ contributes approximately 10% of the total loss, striking a balance between the primary learning objective and the diversification goal. This choice ensures that the diversity term has a meaningful impact without overshadowing the original objective. As suggested by Wabartha et al. (2020), an effective range for $\delta$ is $[10^{-3}, 10^{-1/2}]$, from which we selected $10^{-2}$. While these choices are somewhat arbitrary, they are guided by the intuition that $\mathcal{L}_{\text{div}}^{\text{RL}}$ should be significant enough to influence learning without dominating it. The relative scale of $\lambda$ allows the agent to maintain its focus on task performance while still benefiting from the improved uncertainty estimation provided by diverse critics. Our experiments show that these initial $\lambda$ and $\delta$ values offer a strong baseline, with further fine-tuning yielding additional improvements (see App. B.3 for details).

## 5.1 OVERALL PERFORMANCE

To verify that adding a diversity term does not introduce a negative impact on the overall performance, we assess UARL's performance by tracking cumulative return during training. Fig. 3 shows the

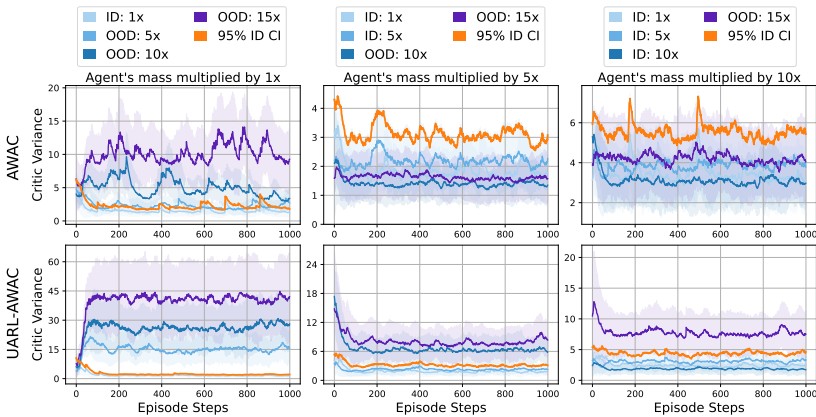

Figure 4: Critic variance across 100 rollouts in the `Ant-v4` environment for AWAC-based methods. The randomized hyperparameter is agent mass. Each column represents a fine-tuning iteration with an expanded ID range by multiplying the agent's mass vector by a constant: $1x \rightarrow 5x \rightarrow 10x$. The orange line indicates the $95\%$ confidence interval of critic variances for ID samples, serving as an OOD detection threshold. UARL-AWAC consistently distinguishes ID from OOD samples, while AWAC struggles to do so.

performance gains in the offline training phase (1st iteration of UARL) across various randomized hyperparameters. We observe significant improvements in the `Ant-v4` and `HalfCheetah-v4`, particularly when randomizing the initial noise scale. For instance, UARL-TD3BC shows a substantial performance advantage over TD3BC in both environments. Similar enhancements are evident when randomizing the friction coefficient, while performance is maintained when altering the agent's mass. Moreover, while EDAC uses an ensemble of 10 critics (Tarasov et al., 2022), UARL achieves strong performance with just 2 critics (default number of critics in baselines), reducing computational overhead. This efficiency makes UARL well-suited for real-world applications with limited resources.

The O2O phase (2nd iteration) expands the range of the randomized hyperparameter, collects new data, balances the replay buffer, and fine-tunes the policy as described in Sec. 4. Fig. 5 showcases the performance during this fine-tuning phase, focusing on AWAC and CQL as our O2O-compatible baselines. The results demonstrate that UARL continues to enhance or maintain performance during the training similar to the 1st phase, and interestingly, it prevents the performance decline observed in CQL across several scenarios, which occurs in `HalfCheetah-v4` when the initial noise scale or friction coefficient is altered. (more details in App. B.1).

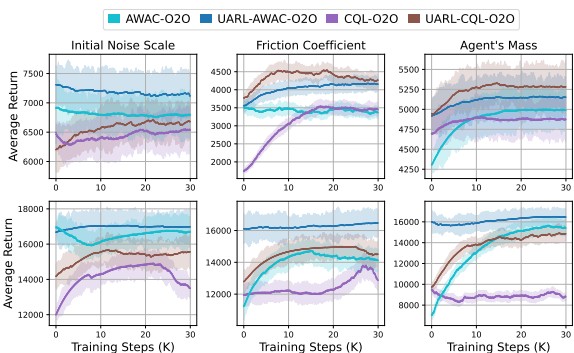

Figure 5: O2O training (2nd iteration) performance in `Ant-v4` (top) and `HalfCheetah-v4` (bottom) environments, showing average return during fine-tuning across three randomized hyperparameters.

### 5.2 OOD DETECTION

A key objective of UARL is to enhance uncertainty awareness in RL policies. We evaluate the variance among critics in UARL and the baselines. To assess OOD detection, we set a threshold based on the 5th percentile of the variances observed in 100 ID rollouts, which refer to rollouts generated in environments with variations the agent has encountered during training. During deployment, data points exceeding this threshold are classified as OOD, allowing us to evaluate the performance on recognizing OOD situations.

Fig. 4 demonstrates the improved uncertainty estimation of UARL-AWAC compared to standard AWAC in the `Ant-v4` environment with randomized agent mass. It shows how UARL enables a clear distinction between ID and OOD states based on critic variance, a capability absent in baseline. Moreover, Fig. 4 illustrates the evolution of uncertainty estimation across three iterations of

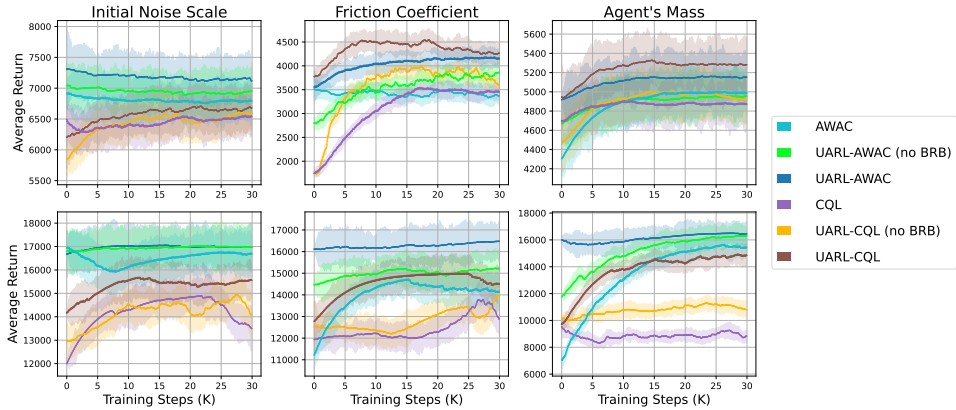

Figure 6: The impact of balancing the replay buffer (BRB) during the initial fine-tuning (O2O) for all randomized hyperparameters. The top row presents results for `Ant-v4`, while the bottom row for `HalfCheetah-v4`.

fine-tuning, each expanding the range of agent mass considered as ID. UARL-AWAC consistently maintains high critic variance for OOD states while adapting its uncertainty estimates as the ID range expands. This adaptive behavior demonstrates UARL's ability to maintain robust OOD detection even as the agent's knowledge of the environment grows, which is crucial for safe real-world RL deployment. Additional comparisons across different environments and parameter settings can be found in App. B.2, while further baseline comparisons are detailed in App. B.5.

## 5.3 EFFECT OF BALANCING REPLAY BUFFER

Finally, we evaluate the impact of the replay buffer balancing mechanism employed in UARL (Alg. 1). Fig. 6 shows the effect of removing the replay buffer balancing feature while fine-tuning the policy in the O2O RL setting. In both `Ant-v4` and `HalfCheetah-v4`, the balancing mechanism consistently outperforms its counterparts across all three randomized hyperparameters. UARL with balancing shows faster learning, higher average returns, and less "unlearning" (overriding of previously learned behaviors), both in `Ant-v4` (especially at the beginning of learning) and `HalfCheetah-v4`. Overall, UARL with balancing demonstrates increasing gains throughout the entire training process.

The results suggest that balancing the replay buffer leads to more stable and efficient learning, particularly crucial in O2O RL where transitioning from offline pretraining to online fine-tuning can be challenging. The consistent improvement across various environmental hyperparameters indicates that the balancing mechanism's benefits are robust to task dynamics variations. In essence, our experiments show that the replay buffer balancing mechanism is a key component in enhancing UARL's performance, accelerating early-stage fine-tuning, and improving overall performance across different environments and task hyperparameters.

## 6 CONCLUSION, LIMITATIONS, & FUTURE WORK

Our proposed pipeline, UARL, addresses real-world RL deployment challenges through targeted, iterative adaptation in simulation. It prevents trial and error via a representative dataset ($\mathcal{D}_w$) from the target environment, deploying only when confident. UARL improves efficiency and robustness with precise OOD detection, balanced O2O RL sampling, and gradual environment variation, without extensive randomization. Its uncertainty estimation is key for detecting OOD scenarios, particularly in robotics, where agents must navigate unexpected variations.

Despite these advantages, UARL has a few drawbacks. As a pilot study, we only explored randomizing a single parameter at the time, while a more elaborate scheme could prove useful for real-world deployment. Moreover, UARL requires a real-world dataset $\mathcal{D}_w$ to determine when to exit Alg. 2. An incomplete or unrepresentative $\mathcal{D}_w$ could lead to overconfident policies. We summarize the limitation here and discuss in detail in App. D.

Future work will focus on automatizing the randomization sequence and validating the robustness of UARL beyond simulated environments, to real robotic systems.

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

# Appendix

## Table of Contents

## A  EXPERIMENT SETUP

### A.1  HYPERPARAMETERS AND NETWORK ARCHITECTURES

As mentioned in Sec. 5, our implementations of baselines and UARL are based on Clean Offline Reinforcement Learning (CORL)[2] (Tarasov et al., 2022). CORL is an Offline RL library that offers concise, high-quality single-file implementations of state-of-the-art algorithms. The results produced using CORL can serve as a benchmark for D4RL tasks, eliminating the need to re-implement or fine-tune existing algorithm hyperparameters. Thus, without tuning any hyperparameter, we use the already provided ones for our experiments, for either baselines or UARL. Following, we present the hyperparameters used in our experiments and the network architectures for baselines.

---

[2]github.com/corl-team/CORL

### A.1.1 AWAC

Table 1: AWAC Hyperparameters.

|  | Hyperparameter | Value |
|---|---|---|
| AWAC (Nair et al., 2020) | Scaling of the advantage estimates | 0.33 |
|  | Upper limit on the exponentiated advantage weights | 100 |
| Common | Discount factor $\gamma$ | 0.99 |
|  | Replay buffer capacity | 2M |
|  | Mini-batch size | 256 |
|  | Target update rate $\tau$ | $5 \times 10^{-3}$ |
|  | Policy update frequency | Every 2 updates |
| Optimizer | (Shared) Optimizer | Adam (Kingma & Ba, 2015) |
|  | (Shared) Learning rate | $3 \times 10^{-4}$ |

---

**Pseudocode 1. AWAC Network Details**

**Critic $Q$ Networks:**
▷ AWAC uses 2 critic networks with the same architecture and forward pass.

```
l1 = Linear(state_dim + action_dim, 256)
l2 = Linear(256, 256)
l3 = Linear(256, 256)
l4 = Linear(256, 1)
```

**Critic $Q$ Forward Pass:**

```
input = concatenate([state, action])
x = ReLU(l1(input))
x = ReLU(l2(x))
x = ReLU(l3(x))
value = l4(x)
```

---

**Policy $\pi$ Network (Actor):**

```
l1 = Linear(state_dim, 256)
l2 = Linear(256, 256)
l3 = Linear(256, 256)
l4 = Linear(256, action_dim)
```

**Policy $\pi$ Forward Pass:**

```
x = ReLU(l1(state))
x = ReLU(l2(x))
x = ReLU(l3(x))
mean = l4(x)
log_std = self._log_std.clip(-20, 2)
action_dist = Normal(mean, exp(log_std))
action = action_dist.rsample().clamp(min_action, max_action)
```

### A.1.2 CQL

Table 2: CQL Hyperparameters.

| | Hyperparameter | Value |
|---|---|---|
| CQL (Kumar et al., 2020) | Scaling the CQL penalty | 1 |
| | Target Action Gap | $-1$ |
| | Temperature Parameter | 1 |
| Common | Discount factor $\gamma$ | 0.99 |
| | Replay buffer capacity | 2M |
| | Mini-batch size | 256 |
| | Target update rate $\tau$ | $5 \times 10^{-3}$ |
| | Policy update frequency | Every 2 updates |
| Optimizer | (Shared) Optimizer | Adam (Kingma & Ba, 2015) |
| | (Shared) Policy learning rate | $3 \times 10^{-5}$ |
| | (Shared) Critic learning rate | $3 \times 10^{-4}$ |

---

**Pseudocode 2. CQL Network Details**

**Critic $Q$ Networks:**
▷ CQL uses 2 critic networks with the same architecture and forward pass.

```
l1 = Linear(state_dim + action_dim, 256)
l2 = Linear(256, 256)
l3 = Linear(256, 1)
```

**Critic $Q$ Forward Pass:**

```
input = concatenate([state, action])
x = ReLU(l1(input))
x = ReLU(l2(x))
value = l3(x)
```

---

**Policy $\pi$ Network (Actor):**

```
l1 = Linear(state_dim, 256)
l2 = Linear(256, 256)
l3 = Linear(256, 256)
l4 = Linear(256, 2 * action_dim)
```

**Policy $\pi$ Forward Pass:**

```
x = ReLU(l1(state))
x = ReLU(l2(x))
x = ReLU(l3(x))
x = l4(x)
mean, log_std = torch.split(x, action_dim, dim=-1)
normal = Normal(mean, std)
action_dist = TransformedDistribution(normal, TanhTransform(cache_size=1))
action = action_dist.rsample()
```

### A.1.3   TD3BC

Table 3: TD3BC Hyperparameters.

|  | Hyperparameter | Value |
|---|---|---|
| TD3BC (Fujimoto & Gu, 2021) | Scaling factor ($\alpha$) | 2.5 |
|  | Noise added to the policy's action | 0.2 |
|  | Maximum magnitude of noise added to actions | 0.5 |
| Common | Discount factor $\gamma$ | 0.99 |
|  | Replay buffer capacity | 2M |
|  | Mini-batch size | 256 |
|  | Target update rate $\tau$ | $5 \times 10^{-3}$ |
|  | Policy update frequency | Every 2 updates |
| Optimizer | (Shared) Optimizer | Adam (Kingma & Ba, 2015) |
|  | (Shared) Learning rate | $3 \times 10^{-4}$ |

---

**Pseudocode 3. TD3BC Network Details**

**Critic $Q$ Networks:**
▷ TD3BC uses 2 critic networks with the same architecture and forward pass.

```
l1 = Linear(state_dim + action_dim, 256)
l2 = Linear(256, 256)
l3 = Linear(256, 1)
```

**Critic $Q$ Forward Pass:**

```
input = concatenate([state, action])
x = ReLU(l1(input))
x = ReLU(l2(x))
value = l3(x)
```

---

**Policy $\pi$ Network (Actor):**

```
l1 = Linear(state_dim, 256)
l2 = Linear(256, 256)
l3 = Linear(256, action_dim)
```

**Policy $\pi$ Forward Pass:**

```
x = ReLU(l1(state))
x = ReLU(l2(x))
x = Tanh(l3(x))
action = max_action * x
```

### A.1.4 EDAC

Table 4: EDAC Hyperparameters.

|  | Hyperparameter | Value |
| --- | --- | --- |
| EDAC (An et al., 2021) | Diversity coefficient $\eta$ | 1.0 |
|  | Target entropy | $-\text{action\_dim}$ |
| Common | Discount factor $\gamma$ | 0.99 |
|  | Replay buffer capacity | 2M |
|  | Mini-batch size | 256 |
|  | Target update rate $\tau$ | $5 \times 10^{-3}$ |
|  | Policy update frequency | Every 2 updates |
| Optimizer | (Shared) Optimizer | Adam (Kingma & Ba, 2015) |
|  | (Shared) Learning rate | $3 \times 10^{-4}$ |

---

**Pseudocode 4. EDAC Network Details**

**Critic $Q$ Networks:**
▷ EDAC uses 10 critic networks with the same architecture.

```
l1 = Linear(state_dim + action_dim, 256)
l2 = Linear(256, 256)
l3 = Linear(256, 256)
l4 = Linear(256, 1)
```

**Critic $Q$ Forward Pass:**

```
input = concatenate([state, action])
x = ReLU(l1(input))
x = ReLU(l2(x))
x = ReLU(l3(x))
value = l4(x)
```

---

**Policy $\pi$ Network (Actor):**

```
l1 = Linear(state_dim, 256)
l2 = Linear(256, 256)
l3 = Linear(256, 256)
mu = Linear(256, action_dim)
log_sigma = Linear(256, action_dim)
```

**Policy $\pi$ Forward Pass:**

```
x = ReLU(l1(state))
x = ReLU(l2(x))
hidden = l3(x)
mu, log_sigma = mu(hidden), log_sigma(hidden)
log_sigma = clip(log_sigma, -5, 2)
policy_dist = Normal(mu, exp(log_sigma))
action = policy_dist.sample()
```

Table 5: Randomized hyperparameter scales used during our experiments. $\rightarrow$ shows one round of fine-tuning (iteration) using UARL, i.e. $E_0 \rightarrow E_1 \rightarrow \cdots \rightarrow E_n$.

| Environment | Randomized Hyperparameter | Original Scale | Modified Scale |
|---|---|---|---|
| Ant-v4 | Initial Noise Scale | $1 \times 10^{-1}$ | $1 \times 10^{-5} \rightarrow 1 \times 10^{-3} \rightarrow 1 \times 10^{-1}$ |
| | Friction Coefficient | 1 | $1 \rightarrow 1.25 \rightarrow 1.5 \rightarrow 1.75$ |
| | Agent's Mass | 1x | $1x \rightarrow 5x \rightarrow 10x \rightarrow 15x$ |
| HalfCheetah-v4 | Initial Noise Scale | $1 \times 10^{-1}$ | $1 \times 10^{-5} \rightarrow 1 \times 10^{-3} \rightarrow 1 \times 10^{-1}$ |
| | Friction Coefficient | 0.4 | $0.4 \rightarrow 0.5 \rightarrow 0.6 \rightarrow 0.7$ |
| | Agent's Mass | 1x | $1x \rightarrow 1.05x \rightarrow 1.1x \rightarrow 1.15x$ |
| Hopper-v4 | Initial Noise Scale | $5 \times 10^{-3}$ | $5 \times 10^{-7} \rightarrow 5 \times 10^{-5} \rightarrow 5 \times 10^{-3} \rightarrow 5 \times 10^{-1}$ |
| | Friction Coefficient | 2 | $2 \rightarrow 2.5 \rightarrow 3 \rightarrow 3.5$ |
| | Agent's Mass | 1x | $1x \rightarrow 1.15x \rightarrow 1.3x \rightarrow 1.45x$ |
| Swimmer-v4 | Initial Noise Scale | $1 \times 10^{-1}$ | $1 \times 10^{-5} \rightarrow 1 \times 10^{-3} \rightarrow 1 \times 10^{-1}$ |
| | Friction Coefficient | 0.1 | $0.1 \rightarrow 0.5 \rightarrow 1 \rightarrow 1.5$ |
| | Agent's Mass | 1x | $1x \rightarrow 5x \rightarrow 10x \rightarrow 15x$ |
| Walker2d-v4 | Initial Noise Scale | $5 \times 10^{-3}$ | $5 \times 10^{-7} \rightarrow 5 \times 10^{-5} \rightarrow 5 \times 10^{-3} \rightarrow 5 \times 10^{-1}$ |
| | Friction Coefficient | 0.9 | $0.9 \rightarrow 2 \rightarrow 3 \rightarrow 4$ |
| | Agent's Mass | 1x | $1x \rightarrow 1.1x \rightarrow 1.2x \rightarrow 1.3x$ |

## A.2 RANDOMIZED HYPERPARAMETER SCALES

Due to the variety of dynamics and physics of the agent in each environment, the range of randomized hyperparameters should be different. For instance, when initializing the Ant-v4 environment, the initial noise scale is $1 \times 10^{-1}$, while it is $5 \times 10^{-3}$ for Hopper-v4. Because of this, we consider various ranges to scale the randomized hyperparameters. The values provided in Table 5 show the scales of the hyperparameters during each iteration of our algorithm.

The selection of these hyperparameter ranges is based on careful consideration of each environment's characteristics and the MuJoCo physics engine's properties (Todorov et al., 2012):

- Initial Noise Scale: This hyperparameter affects the initial state variability. For more stable agents like Ant-v4 and HalfCheetah-v4, we start with a smaller scale ($1 \times 10^{-5}$) and gradually increase it to the default value ($1 \times 10^{-1}$). For less stable agents like Hopper-v4 and Walker2d-v4, we begin with an even smaller scale ($5 \times 10^{-7}$) to ensure initial stability.

- Friction Coefficient: In our experiments, we specifically modify the friction coefficient between the agent and the ground. MuJoCo utilizes a pyramidal friction cone approximation, where this coefficient directly affects contact dynamics and determines how the agent interacts with its environment. We maintain the default friction values initially, then gradually increase them to challenge the agent's locomotion and stability. For Ant-v4, we make moderate increases due to its quadrupedal locomotion's reliance on ground contact. For HalfCheetah-v4, where smooth forward motion is key, smaller increments are used. Higher initial friction coefficients are assigned to Hopper-v4 and Walker2d-v4 to stabilize their balance, and these are increased substantially to test the agents under more challenging conditions.

- Agent's Mass: In MuJoCo, an agent's mass is determined by its constituent geoms. We scale the mass of all geoms uniformly to maintain the agent's mass distribution. For Ant-v4 and Swimmer-v4, we use larger mass increments (up to 15x) as these agents are inherently more stable due to their multi-limbed or water-based nature. For bipedal agents like HalfCheetah-v4, Hopper-v4, and Walker2d-v4, we use smaller increments to avoid drastically altering their delicate balance dynamics.

These hyperparameter ranges are designed to gradually create an OOD scenario while maintaining feasible locomotion, allowing our UARL approach to adapt progressively to more challenging scenarios.

# B    ABLATION STUDY

This section presents a comprehensive ablation study to evaluate the efficacy and robustness of our proposed UARL method. We conduct a series of experiments across multiple MuJoCo environments: `Ant-v4`, `HalfCheetah-v4`, `Hopper-v4`, `Swimmer-v4`, and `Walker2d-v4`. The study applies UARL to three offline RL algorithms: Conservative Q-Learning (CQL), Advantage-Weighted Actor-Critic (AWAC), and TD3BC. We assess the impact of key randomized hyperparameters, namely initial noise scale, friction coefficient, and agent's mass, which are crucial for simulating real-world variability and testing the method's adaptability.

Our evaluation metrics encompass cumulative return during training, OOD detection accuracy, and sample efficiency. We examine these metrics across both the initial offline training phase and the subsequent O2O fine-tuning phases where applicable. To ensure statistical significance and robustness of our findings, each configuration is tested using five random seeds. Throughout the study, we maintain consistency in all hyperparameters, network architectures, and settings, except for those specifically under investigation.

The ablation study is structured to provide insights into several key aspects of UARL. First, we present a detailed analysis of overall performance, expanding on the results provided in the main paper. This includes cumulative return during training for all five environments, offering a comprehensive view of UARL's impact across diverse locomotion tasks.

Next, we delve into the OOD detection capabilities of UARL, a crucial component for safe and robust RL deployment. We examine how the method's uncertainty estimation, implemented through diverse critics, enables effective differentiation between ID and OOD samples. This analysis is particularly relevant for assessing the method's potential in real-world applications where encountering novel situations is inevitable.

We then focus on sample efficiency, a critical factor in the practicality of RL algorithms. By comparing UARL against baselines trained on the full state space from the outset, we demonstrate how our iterative approach to expanding the state space contributes to more efficient learning.

Lastly, we investigate the impact of the balancing replay buffer mechanism, a key innovation in UARL. This component is designed to manage the transition between offline and online learning effectively, and we present results showing its influence on learning stability and performance.

Throughout this ablation study, we aim to provide a nuanced understanding of UARL's components and their contributions to its overall effectiveness. The results and analyses presented here complement and expand upon the findings in the main paper, offering deeper insights into the method's behavior across a range of environments and conditions.

## B.1 OVERALL PERFORMANCE

This subsection presents an extended analysis of UARL's performance across the three MuJoCo environments missing in the main paper: Hopper-v4, Swimmer-v4, and Walker2d-v4. We evaluate the cumulative return during training for each environment, considering the three randomized hyperparameters: initial noise scale, friction coefficient, and agent's mass. This comprehensive analysis builds upon and extends the results presented in Sec. 5 of the main paper.

Fig. 7 illustrates the cumulative return achieved by each agent during the offline training phase (first iteration) across all environments. The results consistently demonstrate that UARL either improves or maintains the performance of the baseline methods. For instance, in the Swimmer-v4 environment, when the friction coefficient is randomized, UARL significantly enhances the performance of both TD3BC and CQL baselines. Similarly, when randomizing the agent's mass in the same environment, we observe performance improvements across all baseline methods. Notably, there are no instances where the application of UARL leads to a decrease in performance. This robustness is particularly evident in challenging environments like Hopper-v4 and Walker2d-v4, where maintaining stability can be difficult. The consistent performance improvements across diverse environments and randomized hyperparameters underscore the versatility and effectiveness of our approach.

Fig. 8 extends this analysis to the fine-tuning phase (second iteration), focusing on the O2O RL setting. Here, we observe that UARL continues to demonstrate strong performance, often surpassing the baselines. This is particularly evident in the Hopper-v4 environment, where UARL-AWAC shows significant improvements over standard AWAC across all randomized hyperparameters.

Fig. 9 presents the same analyses of the fine-tuning phase, but for the *third iteration* of UARL. This figure demonstrates the continued effectiveness of our approach over multiple iterations. The results show that UARL maintains its performance advantages and, in many cases, further improves upon the gains observed in the second iteration. For instance, in the Walker2d-v4 environment, UARL-CQL exhibits consistently superior performance across all randomized hyperparameters, showcasing the method's ability to leverage accumulated knowledge effectively. In the Swimmer-v4 environment, we observe that UARL-AWAC continues to outperform the baseline AWAC, particularly when randomizing the agent's mass and initial noise scale. These results underscore the stability and long-term benefits of our approach, indicating that the performance improvements are not transient but persist and potentially amplify over multiple iterations of fine-tuning.

The results in all three figures highlight a key strength of UARL: its ability to enhance the performance of existing offline RL algorithms without compromising their core functionalities. This is achieved through the introduction of diverse critics and the balancing replay buffer mechanism, which together provide more robust policy learning and effective management of the O2O transition. Furthermore, the consistent performance across different randomized hyperparameters demonstrates UARL's adaptability to various environmental changes. This adaptability is crucial for real-world RL applications, where the ability to handle unexpected variations in the environment is essential for safe and effective deployment.

In summary, these extended results reinforce and expand upon the findings presented in the main paper. They provide strong evidence for the efficacy of UARL across a wide range of locomotion tasks and environmental conditions, highlighting its potential as a robust and versatile approach for offline and O2O RL.

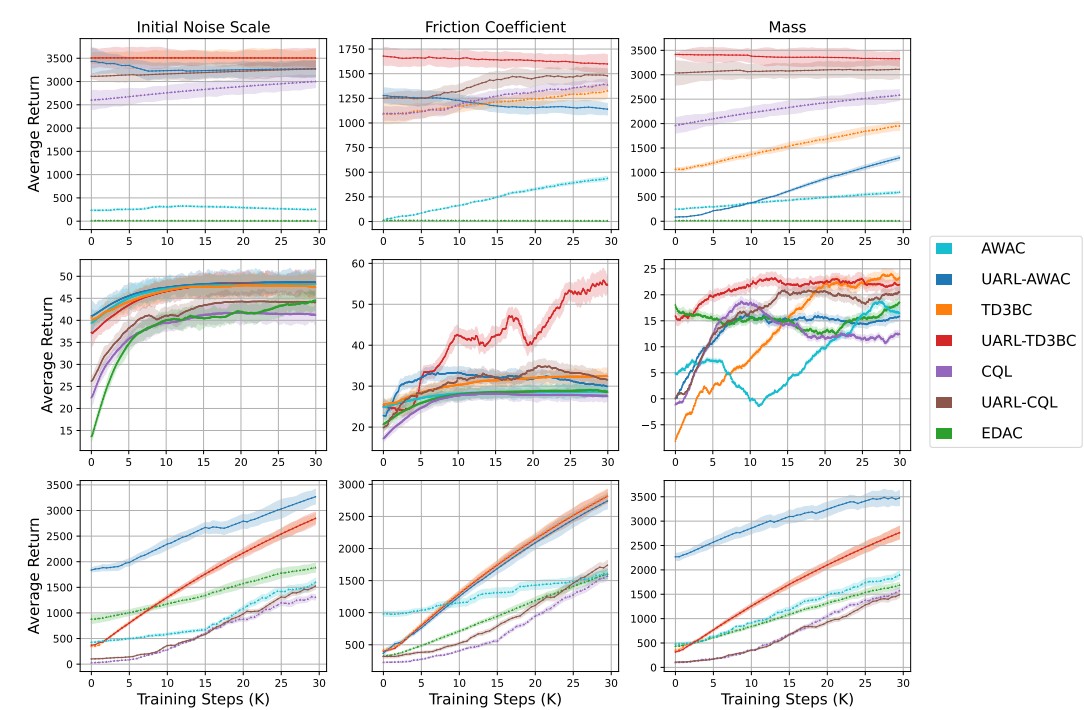

Figure 7: Offline training (first iteration) performance in `Hopper-v4` (top), `Swimmer-v4` (middle), and `Walker2d-v4` (bottom) environments, showing average return during training across three randomized hyperparameters.

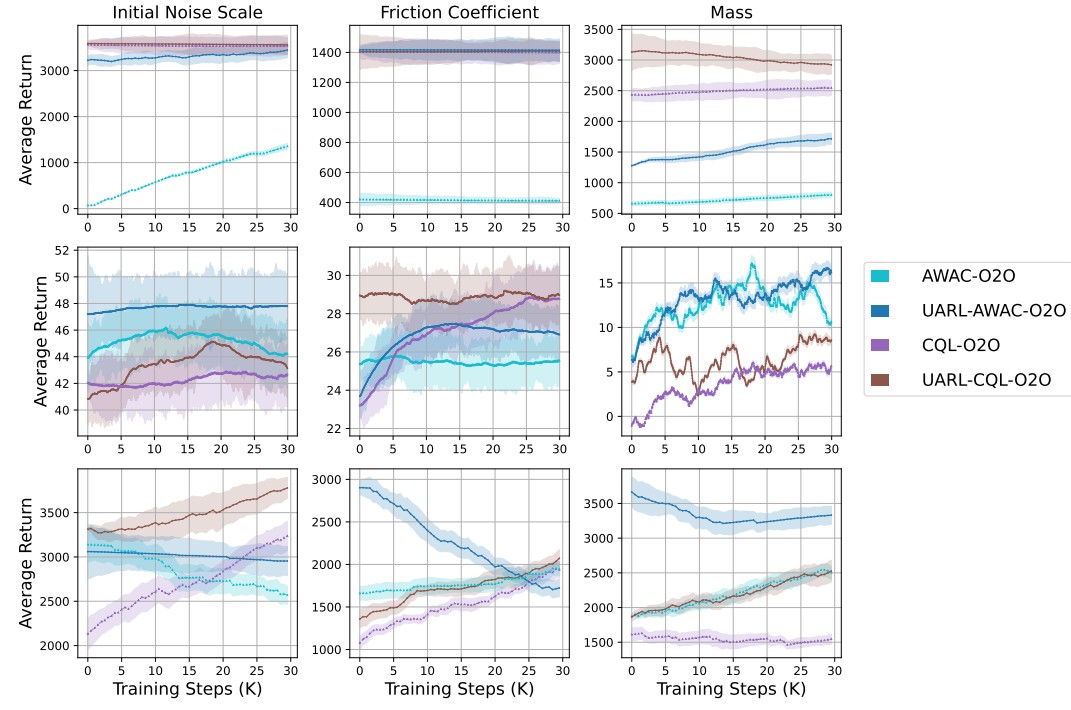

Figure 8: O2O training (second iteration) performance in `Hopper-v4` (top), `Swimmer-v4` (middle), and `Walker2d-v4` (bottom) environments, demonstrating average return during fine-tuning across three randomized hyperparameters.

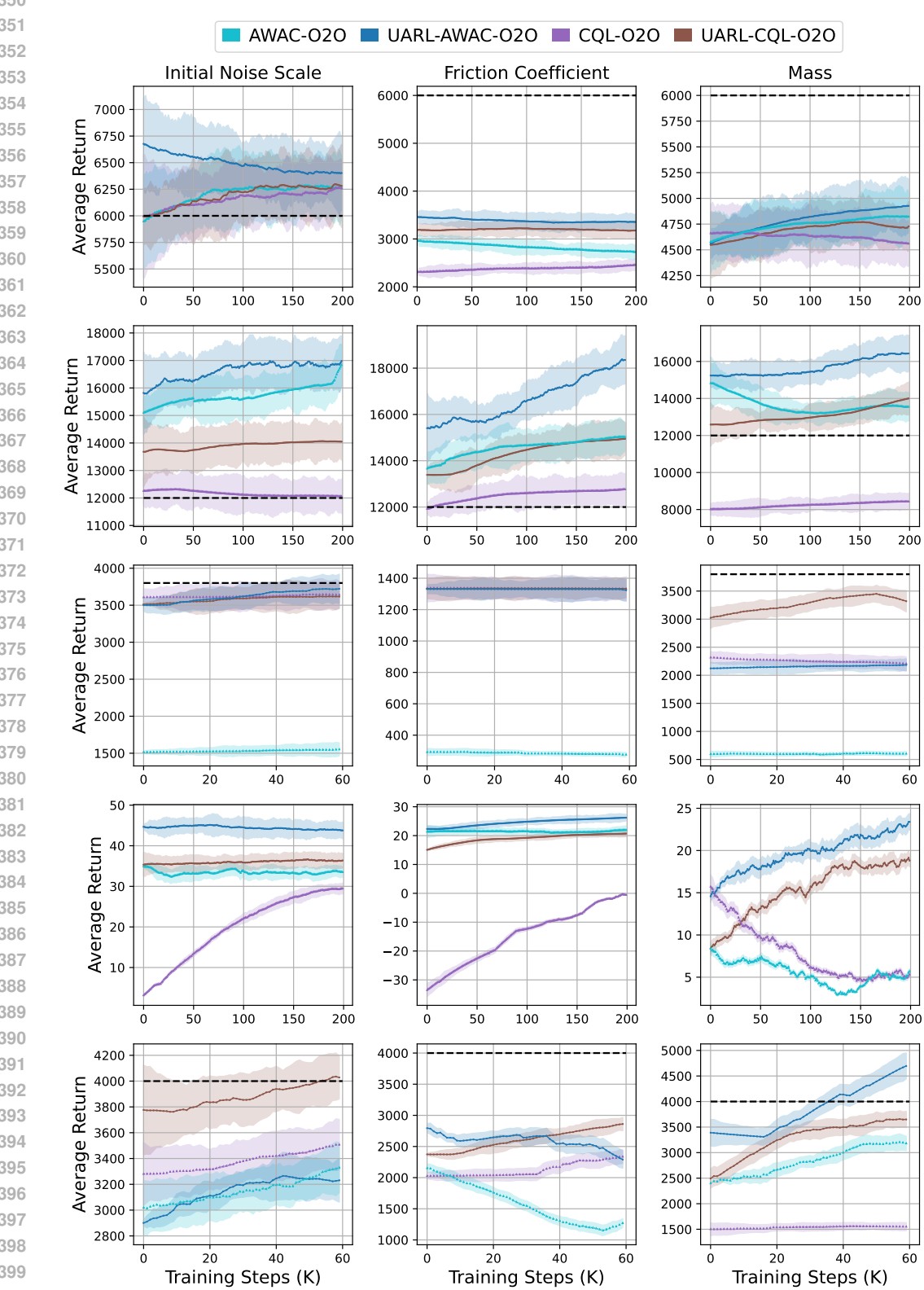

Figure 9: O2O training (third iteration) performance in various environments, demonstrating average return during fine-tuning across three randomized hyperparameters. In order, from top to bottom: Ant-v4, HalfCheetah-v4, Hopper-v4, Swimmer-v4, Walker2d-v4.

## B.2 OOD DETECTION

This section provides an in-depth analysis of UARL's OOD detection capabilities across various environments and randomized hyperparameters. We measure the critic variance across 100 rollouts for both AWAC-based and CQL-based methods, comparing UARL with baseline approaches. In the following figures (Fig. 10 through Fig. 17), each column represents a fine-tuning iteration with an expanded ID range, as detailed in Subsec. A.2. The orange line indicates the 95% confidence interval of critic variances for ID samples, serving as our OOD detection threshold.

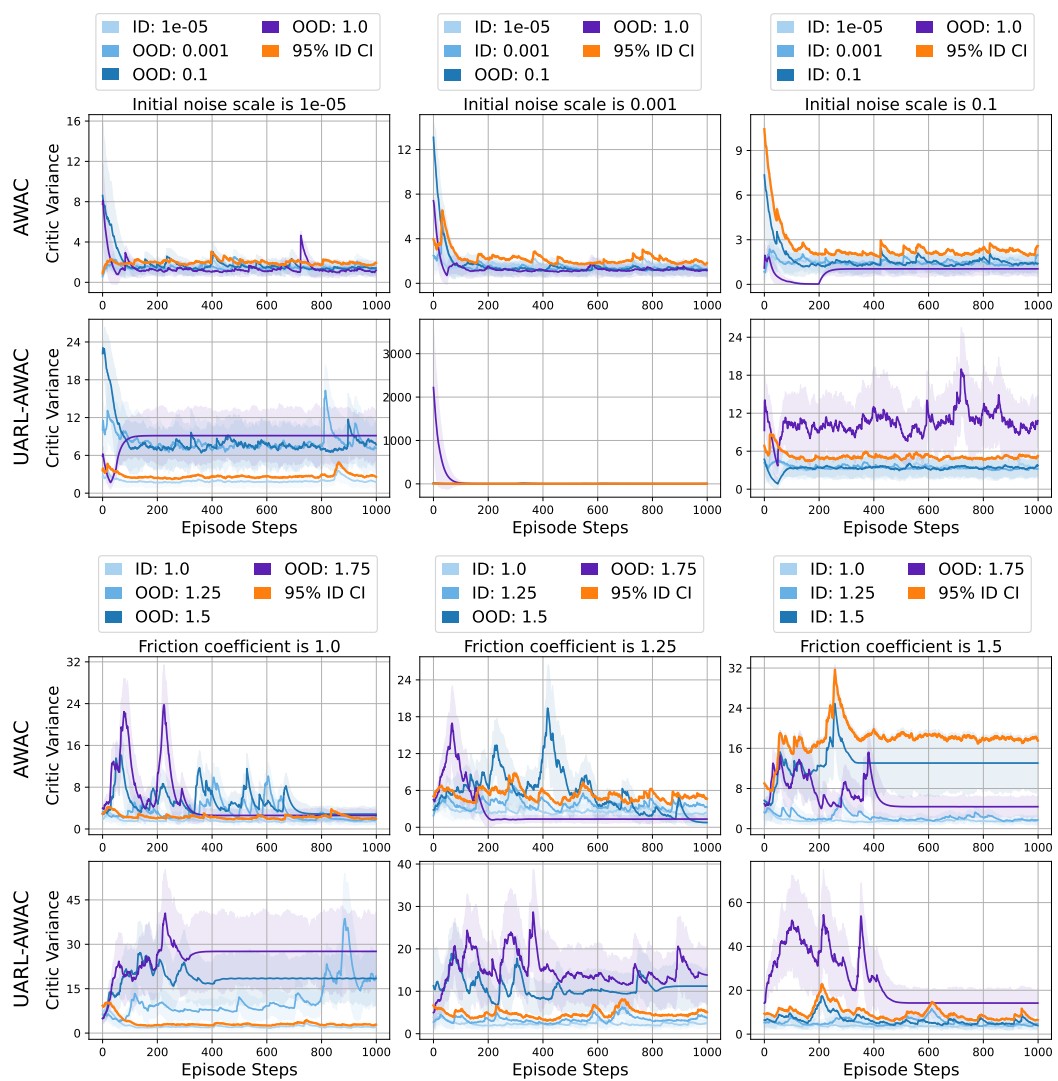

Figure 10: The OOD detection results for AWAC and UARL-AWAC initial noise scale (top) and friction coefficient (bottom) over the `Ant-v4` environment.

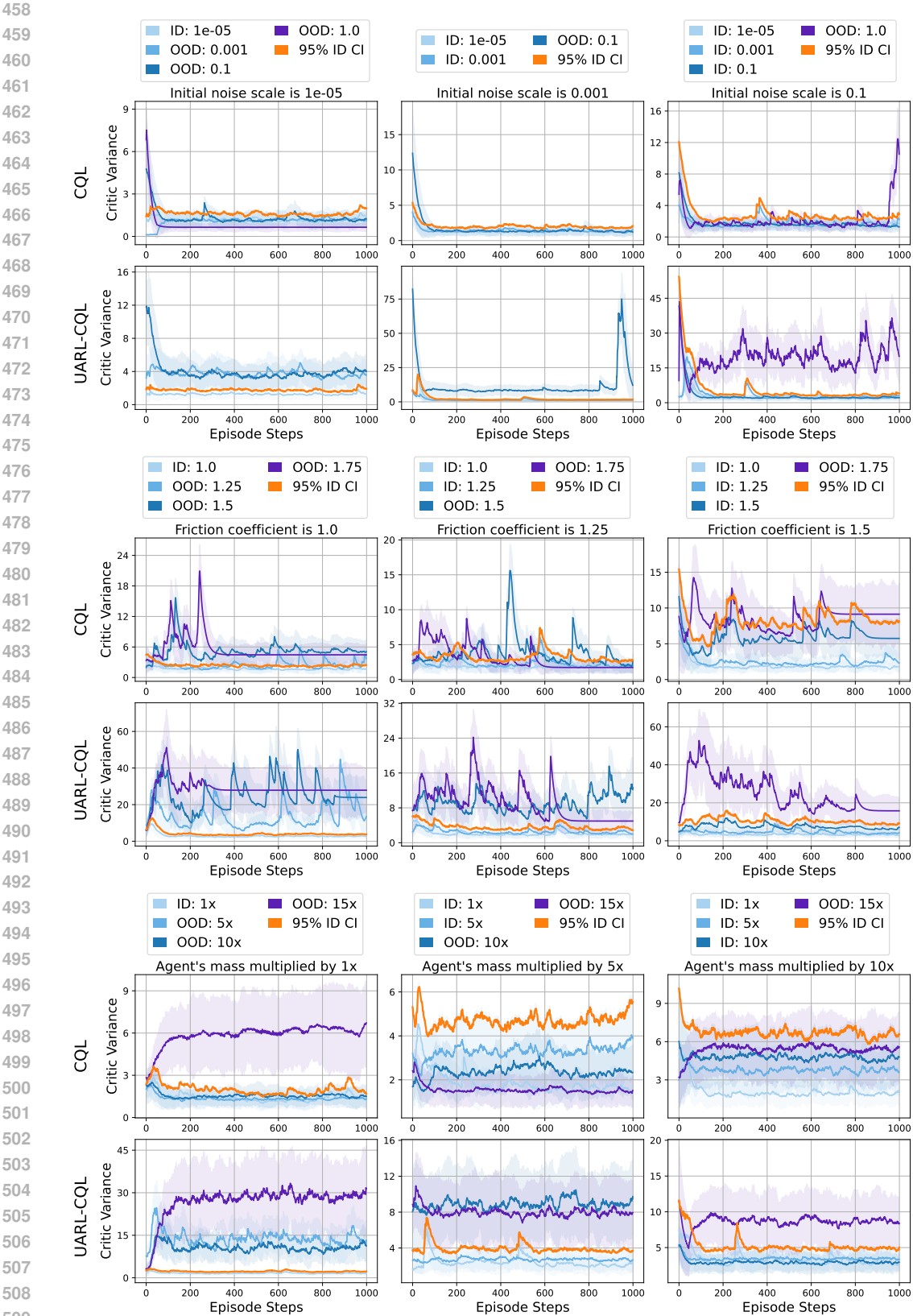

Figure 11: The OOD detection results for CQL and UARL-CQL initial noise scale (top) and friction coefficient (middle), and agent's mass (bottom) over the `Ant-v4` environment.

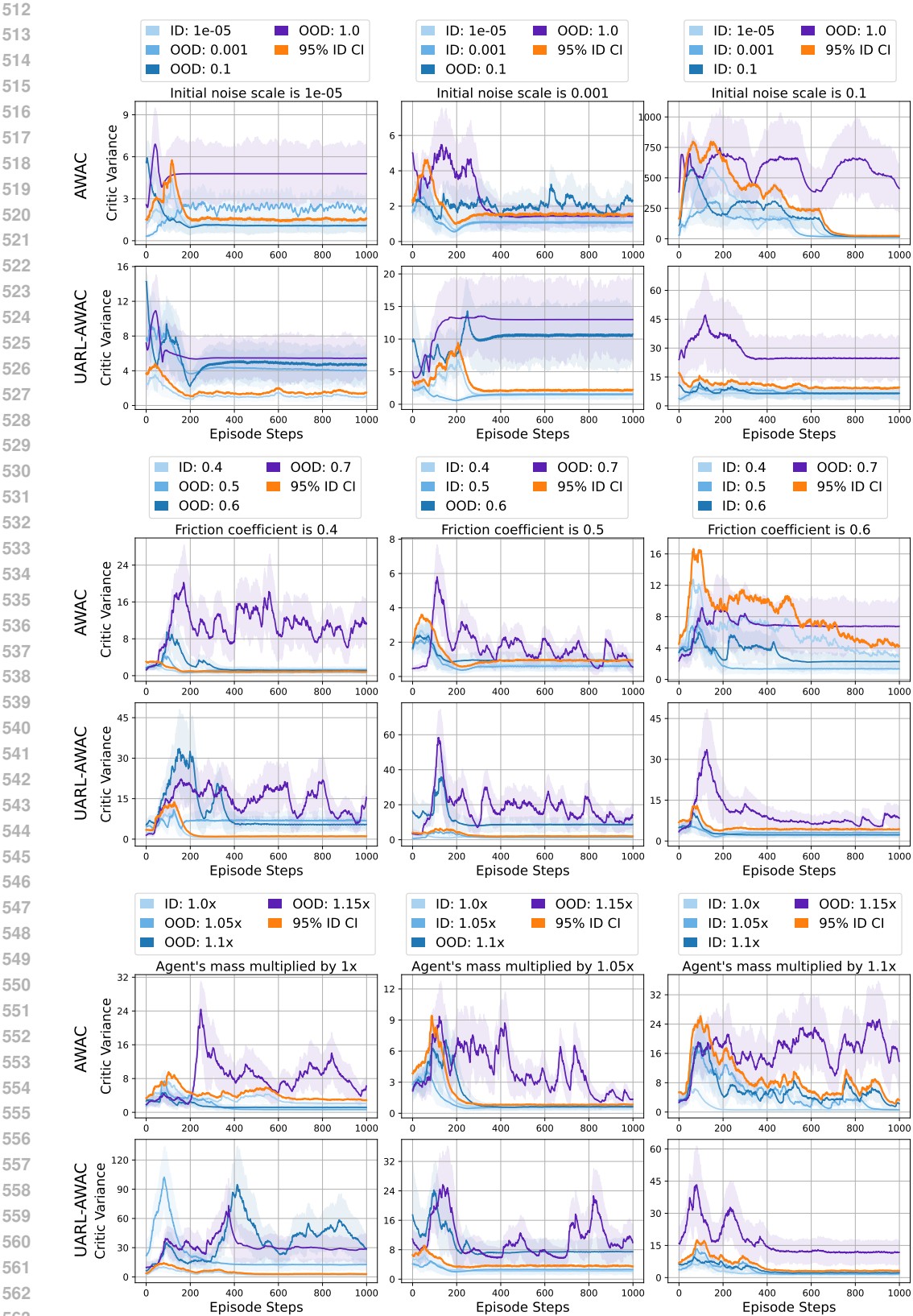

Figure 12: The OOD detection results for AWAC and UARL-AWAC initial noise scale (top) and friction coefficient (middle), and agent's mass (bottom) over the `HalfCheetah-v4` environment.

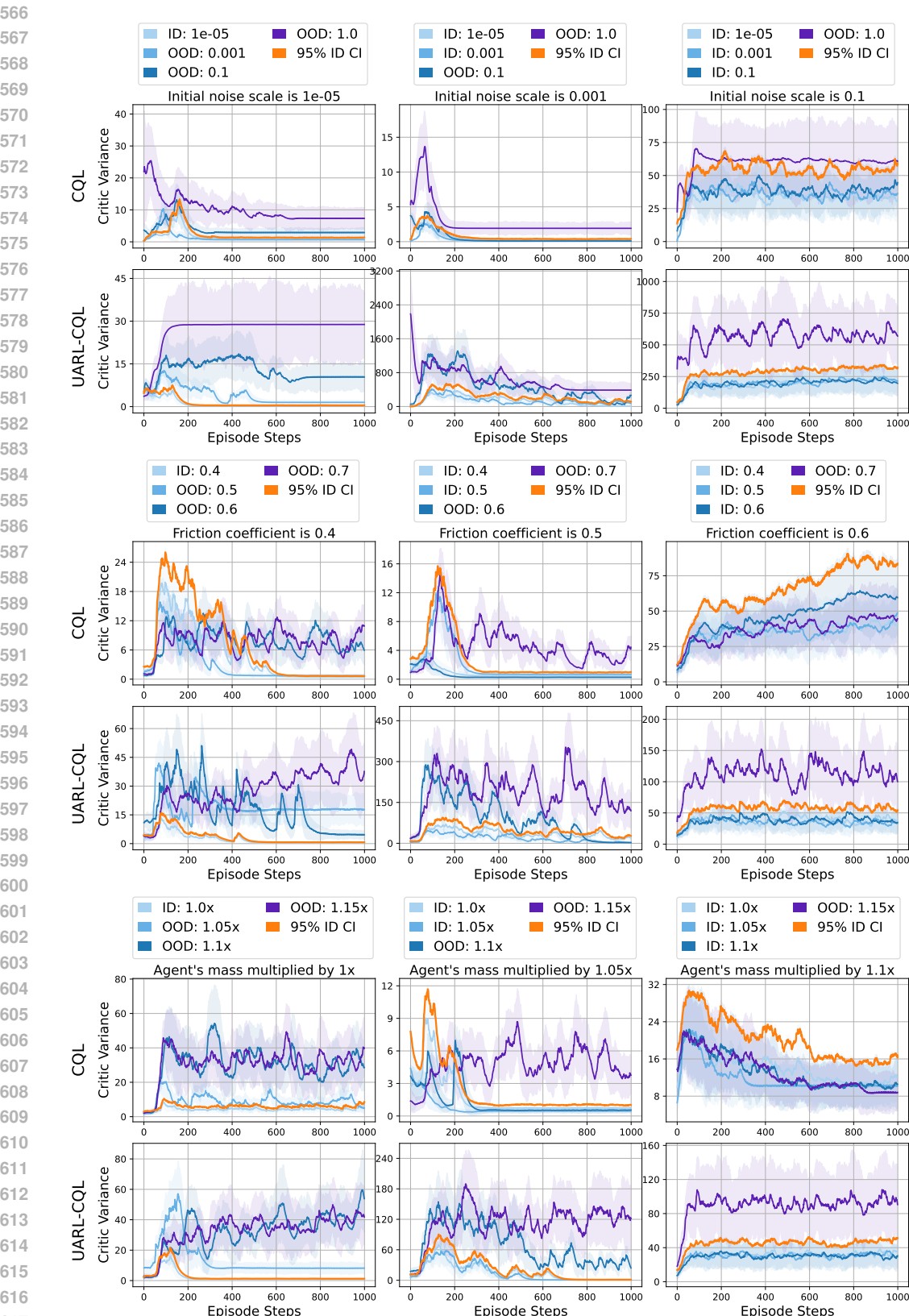

Figure 13: The OOD detection results for CQL and UARL-CQL initial noise scale (top) and friction coefficient (middle), and agent's mass (bottom) over the `HalfCheetah-v4` environment.

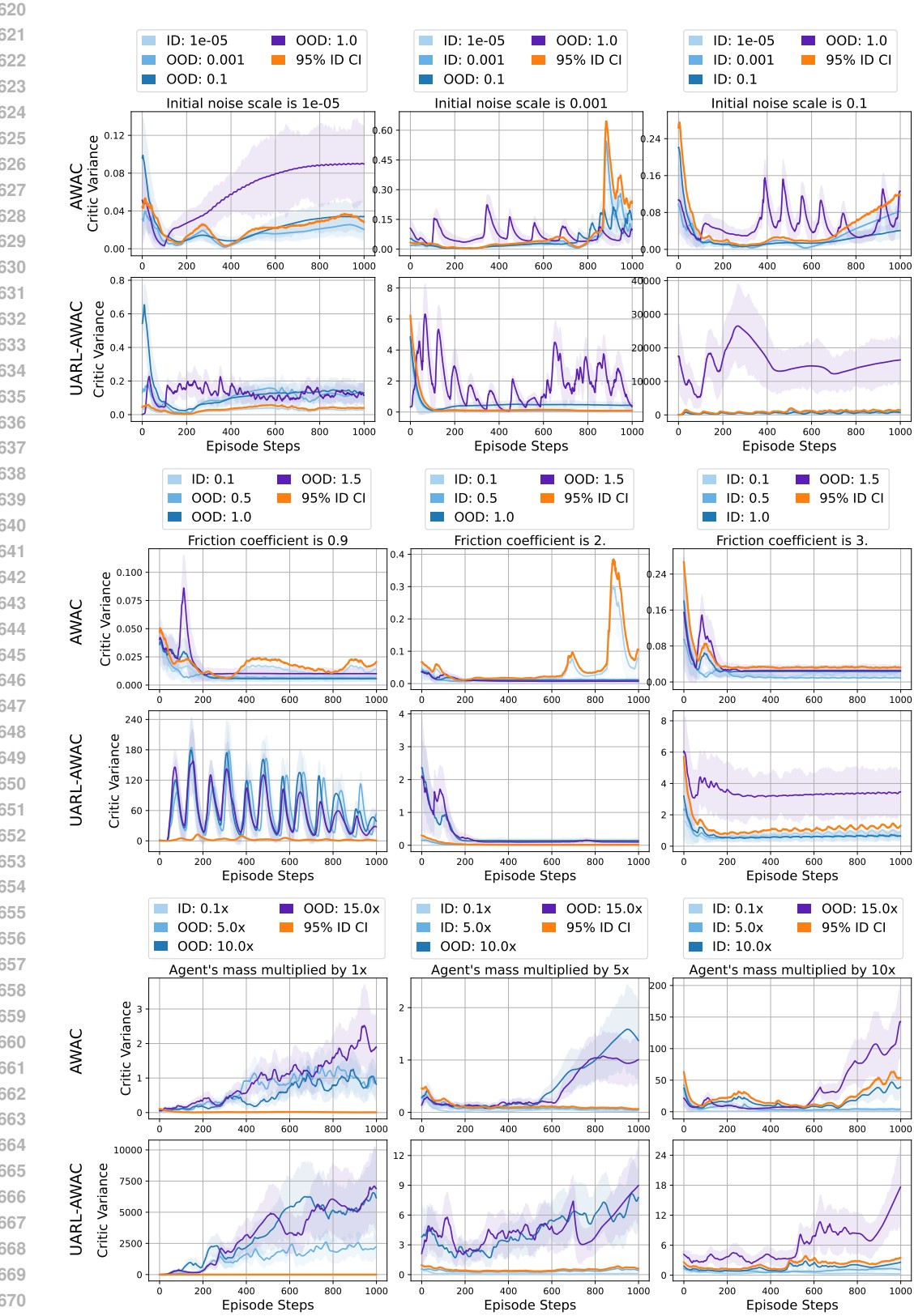

Figure 14: The OOD detection results for AWAC and UARL-AWAC initial noise scale (top) and friction coefficient (middle), and agent's mass (bottom) over the `Swimmer-v4` environment.

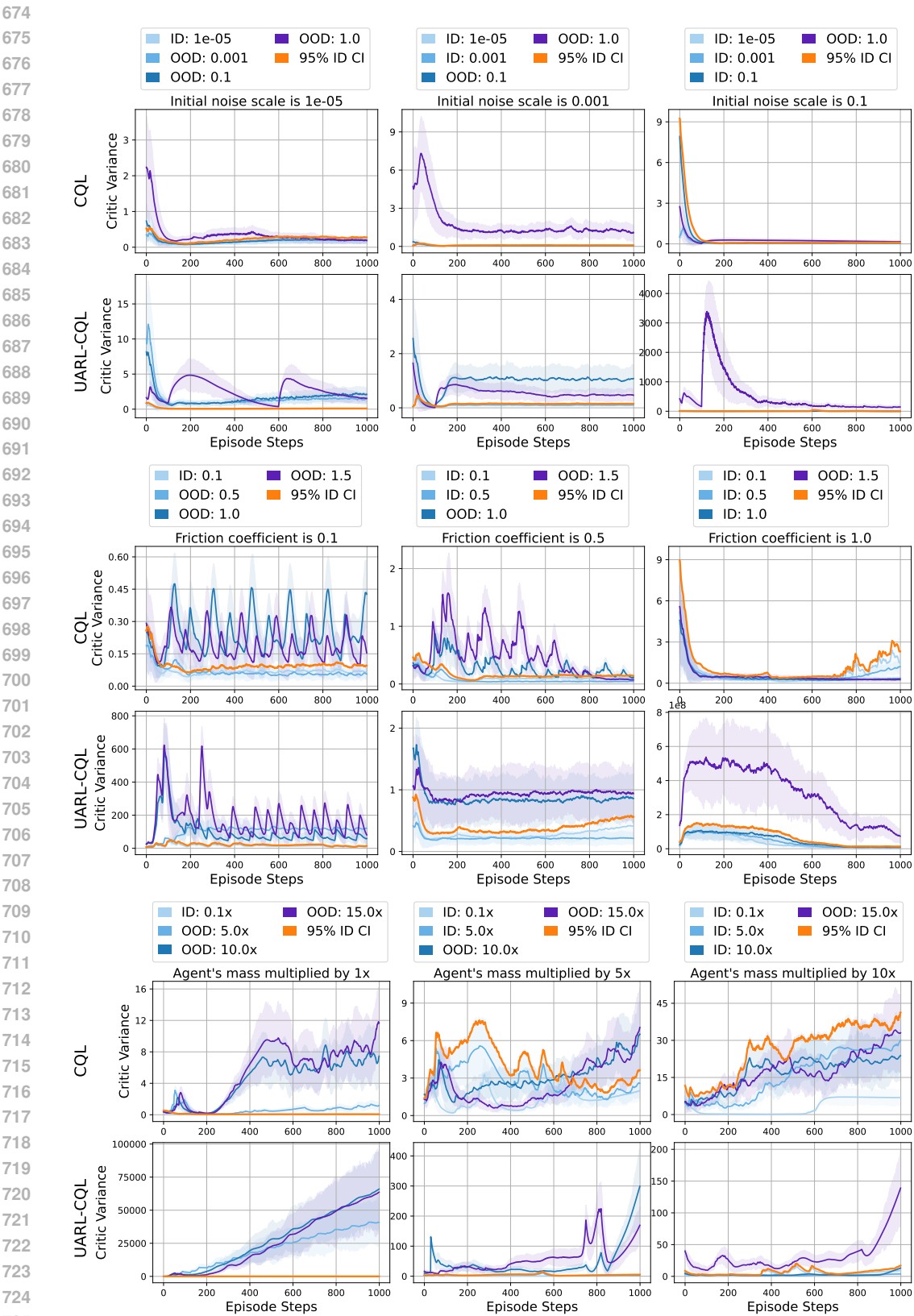

Figure 15: The OOD detection results for CQL and UARL-CQL initial noise scale (top) and friction coefficient (middle), and agent's mass (bottom) over the Swimmer-v4 environment.

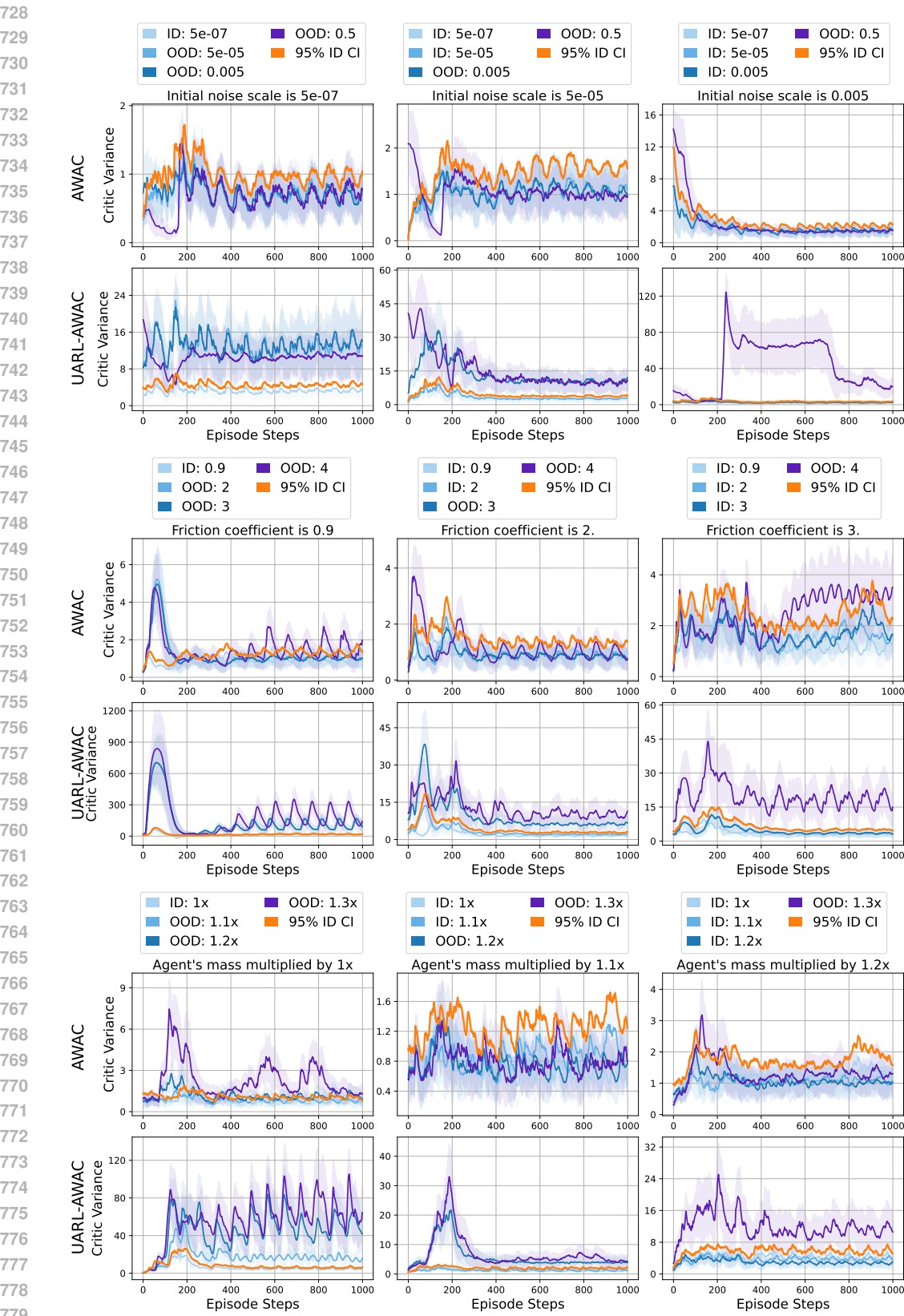

Figure 16: The OOD detection results for AWAC and UARL-AWAC initial noise scale (top) and friction coefficient (middle), and agent's mass (bottom) over the `Walker2d-v4` environment.

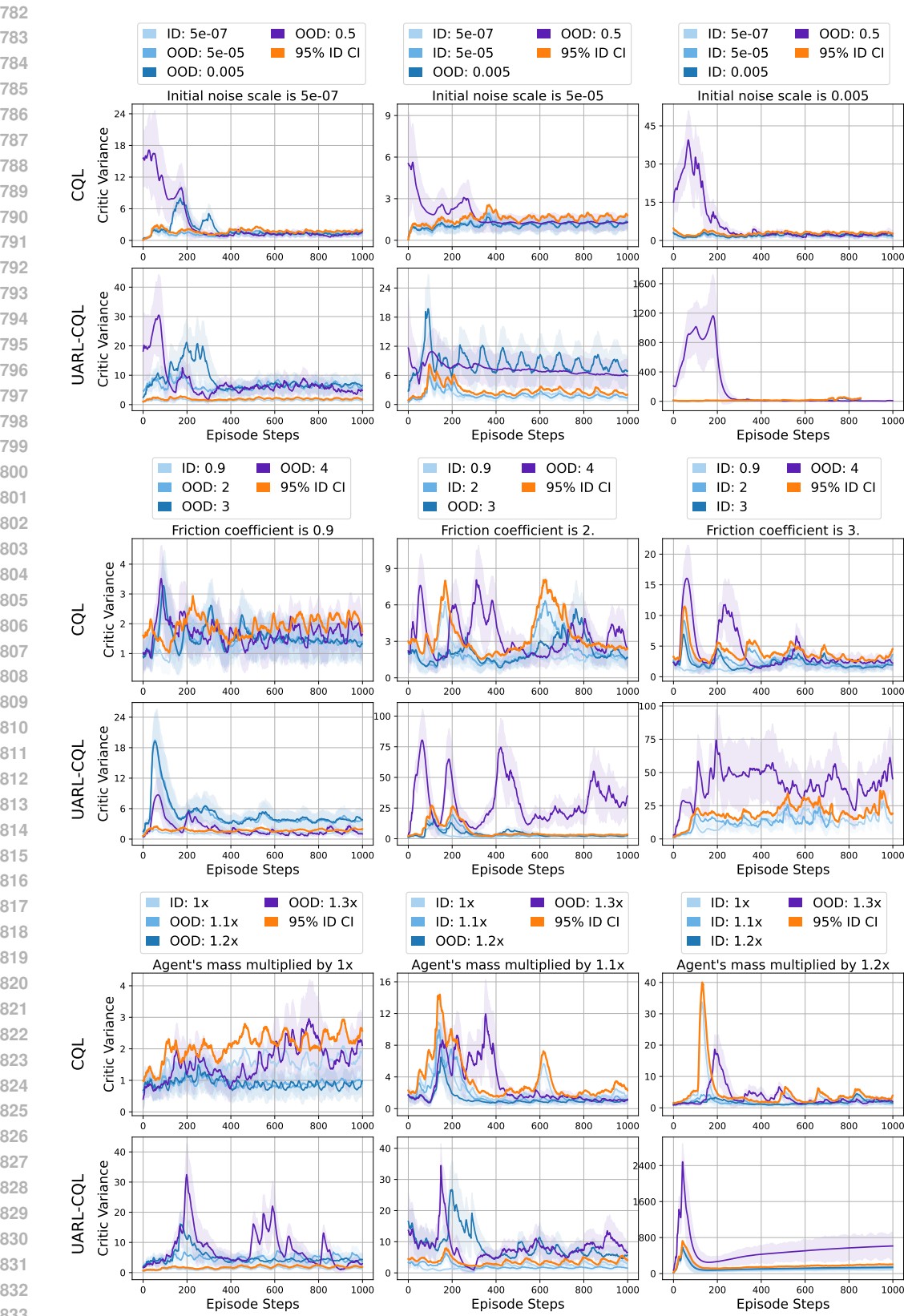

Figure 17: The OOD detection results for CQL and UARL-CQL initial noise scale (top) and friction coefficient (middle), and agent's mass (bottom) over the `Walker2d-v4` environment.

The results show that UARL consistently outperforms baseline methods in detecting OOD data across various environments (`Ant-v4`, `HalfCheetah-v4`, `Swimmer-v4`, and `Walker2d-v4`) using both AWAC-based and CQL-based implementations. A key advantage is its clear separation of ID and OOD samples through critic variance, something often missing in standard AWAC and CQL. UARL adapts its OOD detection threshold as the ID range grows over iterations, maintaining robust performance even in changing environments. Its effectiveness remains consistent across the studied randomized hyperparameters, proving its versatility. The method excels in complex environments like `Ant-v4` and `HalfCheetah-v4` and shows improved or stable OOD detection over time, suggesting it benefits from expanded training data without losing its detection capability. These findings highlight UARL's strong potential for safe, adaptive deployment in dynamic real-world scenarios.

## B.3 UARL Hyperparameter Sensitivity Analysis

This subsection examines the sensitivity of UARL to its key hyperparameters: the diversity coefficient $\lambda$ and the diversity scale $\delta$. These hyperparameters balance the standard RL objective with the goal of promoting diversity among critics. We evaluate various combinations of $\lambda \in \{1\%, 5\%, 10\%, 15\%, 20\%\}$ and $\delta \in \{10^{-3}, 10^{-2}, 10^{-1}\}$ across two settings: Ant-v4, focusing on the agent's mass hyperparameter, and HalfCheetah-v4, focusing on the initial noise scale. We tested these combinations with three baseline algorithms (AWAC, CQL, and TD3BC), using 5 random seeds for each configuration, resulting in a total of 450 experimental runs.

The hyperparameter selection process is grounded in a systematic exploration of the trade-off between diversity regularization and policy optimization. By varying the diversity coefficient $\lambda$ and scale $\delta$, we aim to understand how these parameters influence the learning dynamics of UARL. The chosen ranges reflect a careful consideration of the potential impact of diversity-promoting mechanisms on the RL objective.

Fig. 18 and Fig. 19 demonstrate the influence of varying $\lambda$ and $\delta$ values on agent performance during training, on Ant-v4's mass and HalfCheetah-v4's initial noise scale, respectively. While our chosen *default* configuration performs well, the results indicate the potential for further performance enhancement through careful hyperparameter tuning. The configurations with the most extreme values, specifically $\lambda \in \{15\%, 20\%\}$ and $\delta = 10^{-1}$, tend to have the most detrimental effect on performance. At higher diversity coefficients, the introduced regularization becomes increasingly aggressive, potentially introducing noise that disrupts the learning process. This suggests an inherent trade-off where excessive diversity constraints can impede the algorithm's ability to converge to optimal behavior. This suggests that moderate values for these hyperparameters generally yield better results, with room for fine-tuning within this range to optimize performance for specific environments or baseline algorithms. The observed performance degradation under extreme configurations can be attributed to an amplified diversity loss, which overshadows the RL objective, thereby reducing the UARL's capacity to effectively optimize its policy.

This analysis highlights the robustness of UARL across a range of hyperparameter values while also indicating opportunities for optimization in specific environments or with particular baseline algorithms. The varied performance across different hyperparameter combinations suggests that fine-tuning these hyperparameters could lead to enhanced results in certain scenarios, though the default configuration provides a strong baseline performance across the tested environments and algorithms.

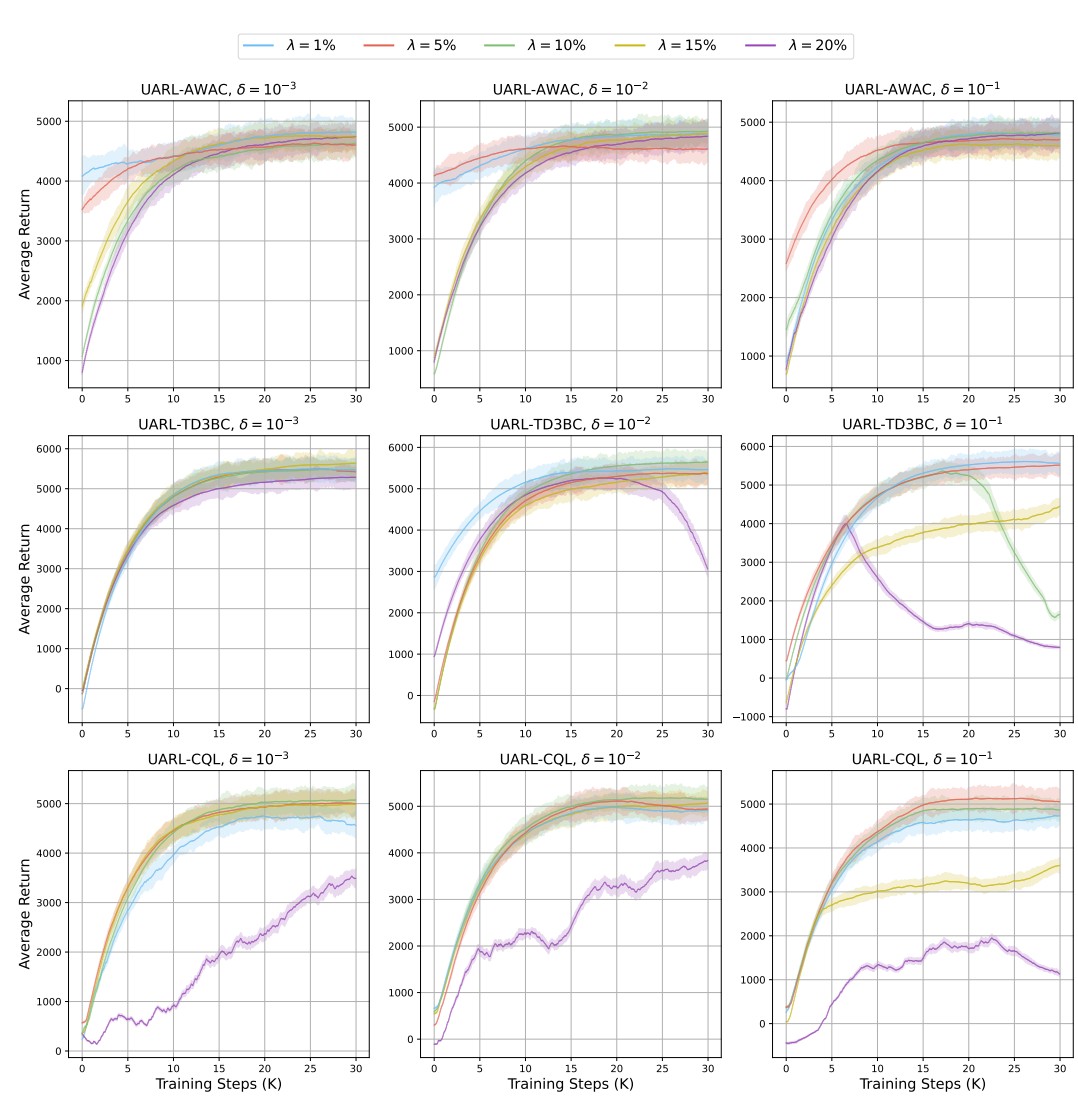

Figure 18: Performance sensitivity to hyperparameters $\lambda$ and $\delta$ during training in the `Ant-v4` environment, with a focus on the agent's mass hyperparameter. Each row represents a baseline algorithm (AWAC, TD3BC, CQL), while each column corresponds to a fixed value of $\delta$ as indicated. The *default* configuration used in the main paper ($\lambda = 10\%$ and $\delta = 10^{-2}$) is depicted in the middle column, highlighted by the green line.

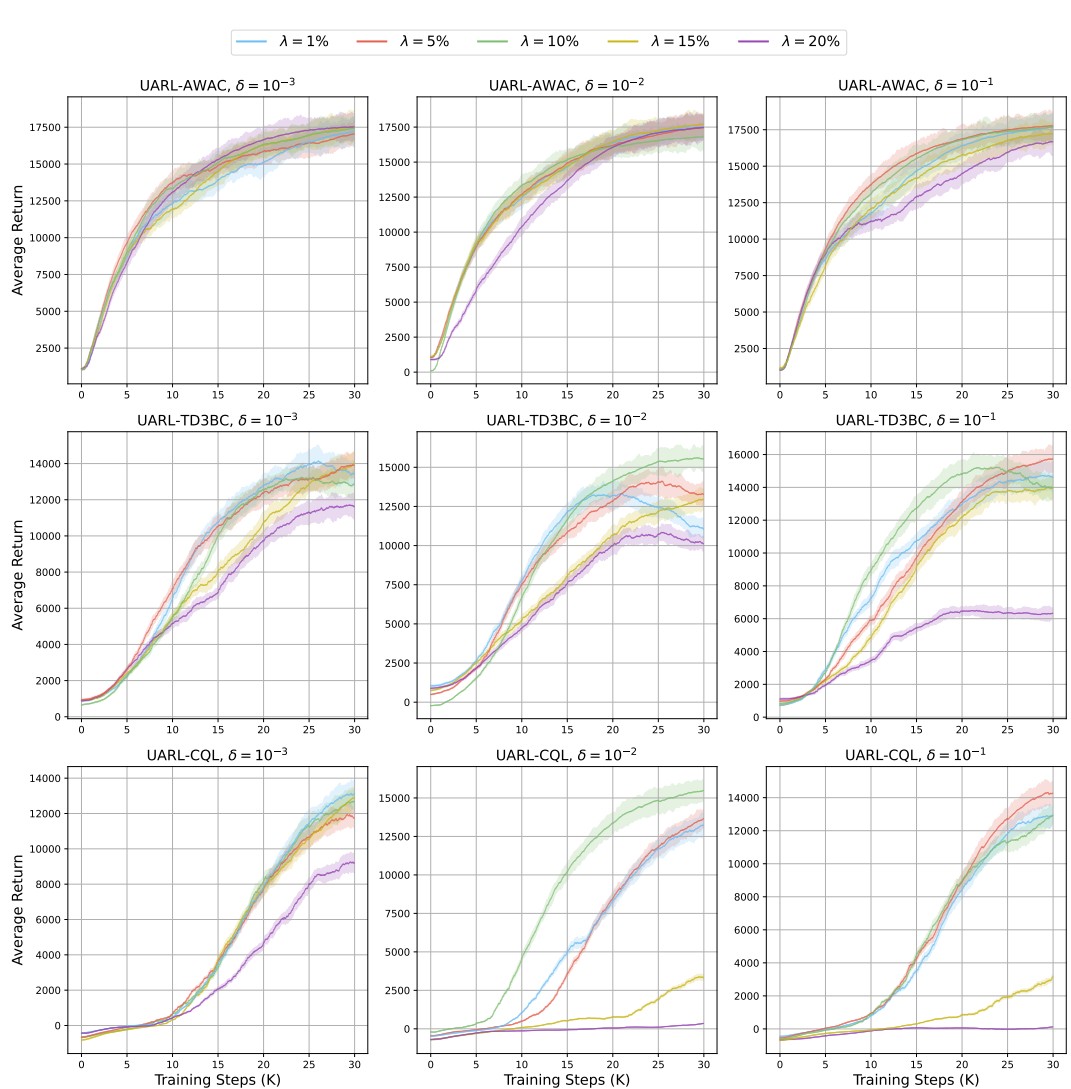

Figure 19: Performance sensitivity to hyperparameters $\lambda$ and $\delta$ during training in the `HalfCheetah-v4` environment, with a focus on the initial noise scale hyperparameter. Each row represents a baseline algorithm (AWAC, TD3BC, CQL), while each column corresponds to a fixed value of $\delta$ as indicated. The *default* configuration used in the main paper ($\lambda = 10\%$ and $\delta = 10^{-2}$) is depicted in the middle column, highlighted by the green line.

## B.4 SAMPLE EFFICIENCY

To assess the sample efficiency of UARL, we define convergence based on the number of data points required for the cumulative return to reach an acceptable level, as per the reward thresholds defined by the Gymnasium Environments (Towers et al., 2023). We trained UARL until convergence, allowing it to progressively expand its state space. For a fair comparison, we then trained baselines on this final expanded state space from the outset, ensuring they operate in the same state space that UARL ultimately reached through its iterative process.

Table 6 compares sample efficiency between UARL and the baselines across different environments and randomized hyperparameters. The figure shows the difference in the number of samples required for convergence. Positive values indicate that baselines require more samples to converge. The results demonstrate that by starting with a limited state space, UARL generally requires fewer samples to converge while maintaining performance as the state space expands in subsequent iterations. Out of a total of 45 comparisons across 5 environments, 3 baselines, and 3 randomized hyperparameters, and based on Welch's t-test with a significance level of 0.05, UARL statistically significantly outperforms the baselines in 35% of cases and achieves non-significant improvements in 49% of cases, resulting in an overall improvement in 84% of comparisons. Notably, a small portion of data from the limited state space (first iteration) is often sufficient for UARL to achieve convergence while remaining performant during the fine-tuning process. This finding supports the notion that iteratively incrementing the state space and fine-tuning the policy significantly enhances sample efficiency.

In calculating UARL's sample efficiency, we considered both the nominal and repulsive datasets to ensure a fair comparison with baselines trained on the expanded (nominal) state space. The results in Table 6 highlight UARL's ability to learn efficiently in complex environments, potentially reducing the computational resources and time required for training robust policies. This sample efficiency, along with the earlier performance improvements, highlights the practical advantages of our approach in real-world RL applications where data collection is costly or time-consuming.

Table 6: Sample efficiency comparison: Difference in means and standard deviations (mean ± std), in thousands of samples needed for convergence (baseline vs. corresponding UARL method: AWAC-based, CQL-based, TD3BC-based). Positive values indicate UARL requires fewer samples.

| Hyperparameter | Ant | HalfCheetah | Hopper | Swimmer | Walker2d |
|---|---|---|---|---|---|
| Initial Noise Scale | $1208_{\pm 859}$ | $8745_{\pm 1499}$ | $5062_{\pm 1986}$ | $5700_{\pm 949}$ | $1680_{\pm 2163}$ |
| | $501_{\pm 1009}$ | $27112_{\pm 1104}$ | $3616_{\pm 1381}$ | $-228_{\pm 899}$ | $1579_{\pm 2440}$ |
| | $5661_{\pm 905}$ | $13799_{\pm 939}$ | $10993_{\pm 1843}$ | $-114_{\pm 837}$ | $-407_{\pm 1488}$ |
| Friction Coefficient | $3857_{\pm 2278}$ | $3765_{\pm 1419}$ | $2202_{\pm 1683}$ | $-43_{\pm 2094}$ | $2357_{\pm 1711}$ |
| | $3578_{\pm 2083}$ | $17343_{\pm 1940}$ | $4038_{\pm 1897}$ | $-376_{\pm 1811}$ | $-930_{\pm 2560}$ |
| | $9401_{\pm 1589}$ | $13043_{\pm 978}$ | $14991_{\pm 1486}$ | $14_{\pm 1736}$ | $-437_{\pm 2102}$ |
| Agent's Mass | $5902_{\pm 1674}$ | $10779_{\pm 1243}$ | $8138_{\pm 1414}$ | $7625_{\pm 2090}$ | $282_{\pm 2217}$ |
| | $8262_{\pm 1540}$ | $1210_{\pm 2220}$ | $2531_{\pm 1941}$ | $-8631_{\pm 2280}$ | $-2646_{\pm 2133}$ |
| | $-1225_{\pm 1120}$ | $13309_{\pm 2915}$ | $2092_{\pm 1387}$ | $3343_{\pm 1762}$ | $1049_{\pm 1549}$ |

### B.5 Comparison with SOTA OOD Baselines

The following experiment evaluates the effectiveness of our method compared to state-of-the-art baselines that focus on robustness and uncertainty estimation. Specifically, we compare our method against PBRL (Bai et al., 2022) and RORL (Yang et al., 2022), which which are designed to enhance robustness, as well as EDAC (An et al., 2021) and DARL (Zhang et al., 2023b), which are focused on improving uncertainty estimation.

PBRL penalizes Q-values in high-uncertainty regions by leveraging the standard deviation of ensemble predictions, promoting conservative learning. To further enhance stability and mitigate extrapolation errors, PBRL incorporates OOD samples into the training buffer. These OOD samples, including states from the offline dataset and actions outside the dataset's support, regularize the Q-function in uncertain regions. By applying stronger penalties for OOD actions in high-uncertainty areas, PBRL achieves robust and reliable policy learning. In contrast, RORL emphasizes robustness against adversarial observation perturbations and OOD states and actions. It employs a conservative smoothing technique to regularize value functions and policies near the dataset's support, penalizing overestimation in unfamiliar regions. RORL further introduces adversarial state perturbations within a predefined radius during training to test and improve robustness. Unlike PBRL, which focuses on OOD actions for ID states, RORL penalizes both OOD states and actions, providing a more comprehensive form of conservatism. Additionally, RORL minimizes policy distribution differences under perturbed states, ensuring stability in adversarial scenarios, making it particularly effective in environments with challenging observation perturbations and limited data coverage.

EDAC aims to prevent over-estimation of Q-values during training with OOD samples, ensuring a conservative learning process. This is achieved by diversifying the gradients of Q-functions in an ensemble, thereby reducing the alignment of Q-function gradients. By minimizing their cosine similarity, EDAC increases the variance in Q-value predictions for OOD actions, which enhances the ability to penalize uncertain or risky actions. This uncertainty-aware approach helps avoid over-confident predictions, leading to more robust offline RL. On the other hand, DARL employs a non-parametric particle-based cross-entropy estimator that uses k-nearest neighbor search to measure uncertainty, projecting data into a distance-preserving, low-dimensional space to make the process efficient. This estimator allows DARL to accurately quantify uncertainty by capturing the relationship between samples in the dataset. Building on this, DARL incorporates adaptive truncated quantile critics, which adjust the extent of underestimation for Q-values based on sample uncertainty, ensuring conservative value estimation for high-uncertainty samples.

Fig. 20 extends the results from the main paper's Fig. 4 by including the performance of PBRL, RORL, EDAC, and DARL in OOD detection. As shown, PBRL and RORL struggle to effectively differentiate between ID and OOD dynamics, leading to overly optimistic policies in high-uncertainty regions and reduced robustness in OOD scenarios. EDAC and DARL demonstrate some success in distinguishing ID from OOD cases, but their performance is inconsistent across settings. For instance, EDAC accurately separates ID and OOD samples during the $1^{st}$ iteration of fine-tuning ($2^{nd}$ column), but achieves only $30\%$ accuracy in distinguishing ID from OOD samples during the $2^{nd}$ iteration. Similarly, DARL performs well during the offline training phase ($1^{st}$ column) but fails to maintain consistent performance thereafter. In contrast, UARL consistently outperforms these baselines, reliably distinguishing ID and OOD dynamics across all settings, resulting in more robust and dependable policy behavior.

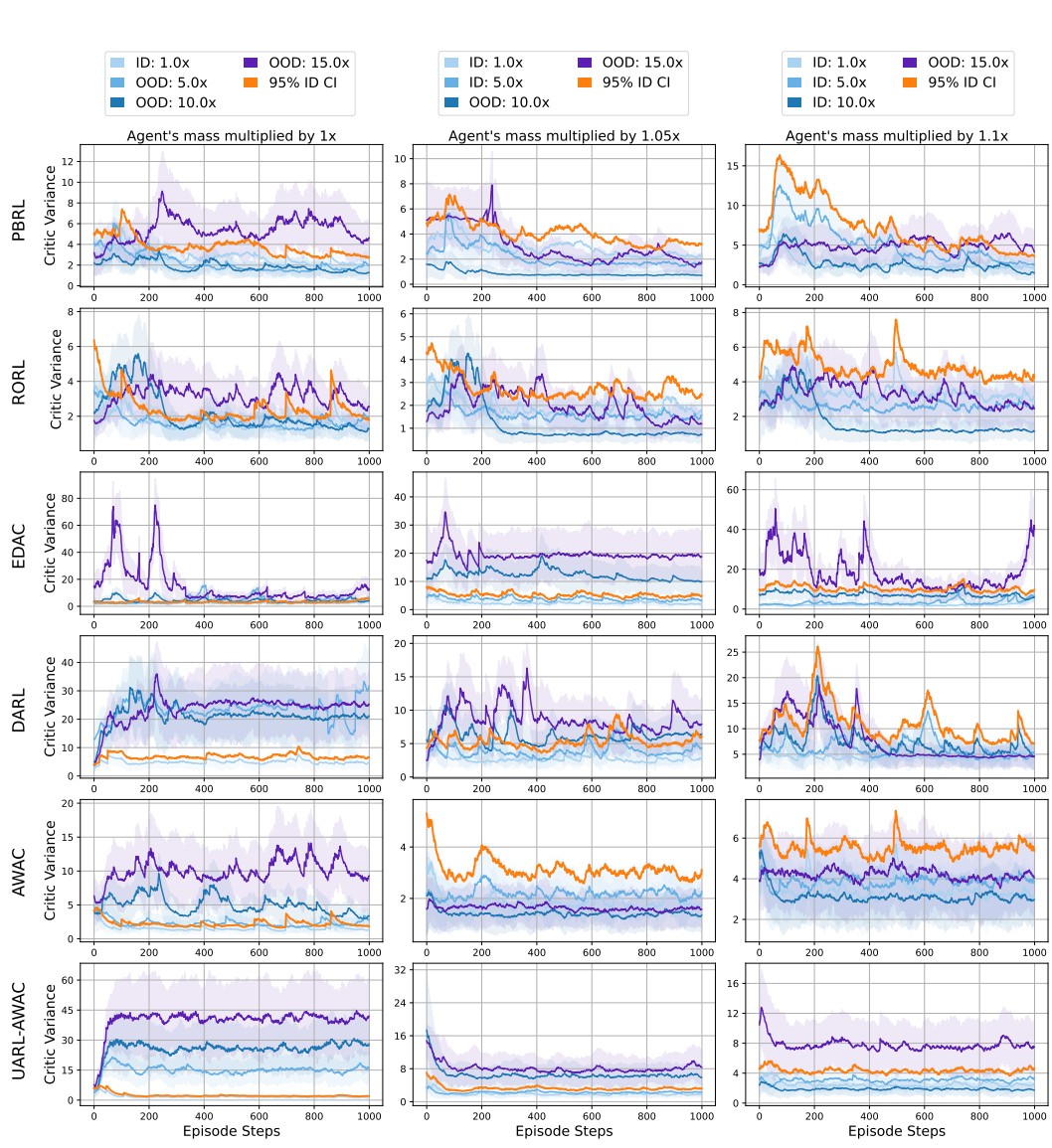

Figure 20: Critic variance across 100 rollouts in the `Ant-v4` environment for PBRL, RORL, EDAC, DARL, and AWAC-based methods. The randomized hyperparameter is agent mass. Each column represents a fine-tuning iteration with an expanded ID range by multiplying the agent's mass vector by a constant: 1x → 5x → 10x. The orange line indicates the 95% confidence interval of critic variances for ID samples, serving as an OOD detection threshold. UARL-AWAC consistently distinguishes ID from OOD samples, while AWAC struggles to do so.

## C    RELATED WORK

**Safe and Robust RL** addresses critical challenges in high-stakes applications like robotics (García & Fernández, 2015). While safe RL focuses on avoiding harmful actions by imposing policy constraints during training and deployment (Heger, 1994), robust RL aims to maintain performance stability under uncertainties or adversarial perturbations (Iyengar, 2005; Nilim & El Ghaoui, 2005). To enhance robustness, domain randomization techniques train agents in varied, randomized environments (Andrychowicz et al., 2020; Tobin et al., 2017; Lee et al., 2020; Mozian et al., 2020). However, this approach can be computationally expensive and challenging to tune for complex scenarios (Mehta et al., 2020). Additionally, those methods assume that policy can be trained in OOD scenarios, which poses significant risks in real-world applications.

Our proposed method, UARL, addresses these challenges by explicitly detecting OOD scenarios and adapting policies based on uncertainty estimation without direct interactions in OOD environments. This approach improves both safety and robustness without incurring the computational overhead associated with traditional domain randomization techniques.

**Uncertainty-aware RL** While there are several uncertainty-aware methods in offline RL, such as MOBILE (Sun et al., 2023), PBRL (Bai et al., 2022), and RORL (Yang et al., 2022), which penalize OOD actions using uncertainty quantifiers, we respectfully distinguish our approach. Unlike MOBILE, PBRL, and RORL, which primarily focus on robustness by penalizing uncertain actions to avoid OOD scenarios during training, our work focuses on explicit OOD detection during deployment. This difference is crucial: while these methods aim to build robust policies that can operate in OOD conditions, our method is designed to identify when a system is operating in OOD situations prior to policy deployment, which is critical in safety-critical systems like robotics.

Our progressive environmental randomization method builds an agent's ability to distinguish between ID and OOD states, actively detecting when novel or uncertain scenarios arise. This capability allows the agent to make informed decisions, such as requesting human intervention or guiding data collection for further policy refinement. While robustness is a beneficial side effect of our iterative fine-tuning, it is secondary to our primary goal of reliable OOD detection.

**Curriculum learning** in RL aims to improve agent learning by progressively exposing the agent to increasingly complex tasks or environments (Narvekar et al., 2020; Li et al., 2024). Traditional approaches involve manually designing a sequence of tasks with increasing difficulty, where agents learn foundational skills before tackling more challenging scenarios (Graves et al., 2017). Recent advances have explored more nuanced transfer methods, such as REvolveR (Liu et al., 2022), which introduces a continuous evolutionary approach to policy transfer by interpolating between source and target robots through a sequence of intermediate configurations. While REvolveR focuses on morphological and kinematic transitions, it still shares the fundamental limitation of most curriculum learning approaches: relying on predefined task progressions that may not capture the unpredictability of real-world environments.

In contrast, our approach focuses on uncertainty-driven adaptation, dynamically expanding the exploration space based on real-time uncertainty estimation rather than a predetermined task hierarchy. Unlike curriculum RL's structured task progression, UARL enables continuous, adaptive learning that more closely mimics real-world environmental variability, particularly in scenarios with unpredictable and out-of-distribution events. Critically, our method avoids the safety risks associated with direct policy refinement in target domains by using an ensemble of critics to evaluate policy suitability without dangerous direct interactions.

Table 7: Computational Overhead of CQL vs. CQL+UARL, on a single Nvidia 4090 GPU.

| Metric | Baseline | UARL |
|---|---|---|
| Memory Usage | ~2 GB | ~4 GB |
| Memory Increase | 50% | |
| Training Time per Iteration | ~0.5 seconds | ~0.55 seconds |
| Computational Time Increase | 10% | |
| Diversity Loss Calculation | Not Applicable | Required |

## D LIMITATIONS

While UARL demonstrates promising results in enhancing safety in RL, several key limitations warrant discussion.

Table 7 highlights the computational overhead associated with UARL. The approach introduces moderate resource demands, with a 10% increase in training time and 50% higher memory usage compared to baseline methods which depends on the size of the repulsive dataset. However, these costs are offset by the potential for improved policy safety and robustness, particularly in detecting and adapting to OOD scenarios.

A primary limitation of UARL is the sequential randomization of a single hyperparameter during iterative adaptation. While this controlled exploration aids stability, it may fail to capture the complex interplay between multiple environmental parameters. Future work could explore simultaneous multi-parameter randomization to better simulate real-world uncertainties.

The method also relies on manually defined parameter ranges for randomization, determined using domain expertise. This reliance may limit generalizability across diverse tasks and environments. Developing an automated mechanism to adaptively determine parameter ranges could enhance scalability and reduce human intervention.

Another challenge is the dependence on a proxy dataset $\mathcal{D}_w$ from the target environment, which critically influences uncertainty estimation and policy refinement. If this dataset is incomplete or unrepresentative, it can lead to suboptimal adaptations or overconfident policies. Techniques to ensure dataset quality and representativeness will be crucial for robust performance.

Finally, while UARL has demonstrated effectiveness on MuJoCo benchmark tasks, its applicability to more complex, high-dimensional, real-world scenarios remains an open question. Extensive validation across diverse robotic and control domains will be necessary to establish its broader relevance and effectiveness.

