# OpenReview forum: "Mitigating Distribution Shifts: Uncertainty-Aware Offline-to-Online Reinforcement Learning"
_ICLR.cc/2025/Conference — Submitted to ICLR 2025_

### Official Review · Reviewer_ruzS · 2024-10-29

**Soundness:** 2
**Presentation:** 3
**Contribution:** 2
**Rating:** 5
**Confidence:** 4

**Summary:**

This paper presents a novel approach aimed at solving the OOD problem faced by deploying reinforcement learning strategies in real-world scenarios when the distribution of training environments is shifted. The proposed approach tackles this issue by adopting a new diversity ensemble Q-network approach to OOD detection. Furthermore, the method incorporate an iterative policy fine-tuning method that starts with offline training in the original environment and gradually scales up to more stochastic environments through online interactions. Experimental results show that this approach outperforms the Baseline algorithm in Mujoco environments with randomized environmental hyperparameter and typically requires fewer samples to converge.

**Strengths:**

1.This paper propose a novel OOD detection method and an iterative online policy fine-tuning training framework.
2.Good experimental results are obtained on Mujoco environments with randomized environmental hyperparameter, verifying the validity of the method.
3.The writing is good.

**Weaknesses:**

1. It should be further shown, either theoretically or experimentally, why the diversity term $\mathcal{L}^{\text{RL}}_{\text{div}} $ in Eq. 5 allows the ensemble Q network to learn the value of diversity. Intuitively, minimising $\text{exp}(-\Vert Q_i(s, a) - (r + Q_i(s^\prime, a^\prime)) \Vert^2/2\delta^2)$ will allow the Q network to not converge quickly to a certain value on the repulsive dataset, but it does not guarantee that the ensemble Q network learns diverse values.
2. Further clarification is needed for how to calculate $V_Q$ in Algorithm 2 and how to calculate critical variance in uncertainty estimation experiments.
3. The ablation experiments in Appendix B.3 are not detailed enough. The training curves for different parameter combinations should be differentiated to illustrate the algorithm's parameter sensitivity to $\lambda$ and $\delta$ during training.
4. For each $E_i$, the randomized environmental hyperparameter range is determined without a common metric but as a hyperparameter, which may require a lot of time for online tuning for complex scenarios.

**Questions:**

1. According to Eq. 5, $R(s, a)$ is not sampled from the dataset $\mathcal{D}$, but the general Q-function update for offline RL, as in Eq. 1, is to use the sampled $r$. Is this a mistake here?
2. Is there a performance or computational advantage of UARL over direct processing of $E_\omega$'s using the Robust RL algorithm or the algorithm with domain randomization techniques? Can this be illustrated experimentally?
3. Notice that in offline training (1st iteration), EDAC performs much worse than CQL and TD3+BC in many environments, which doesn't seem to match the experimental results in the EDAC article?
4. The experiments in this paper were all performed in Mujoco, how do we obtain the real-world demonstration dataset $\mathcal{D}_\omega$ in the simulation environment like Mujoco?

---

> ### Author Response · Authors · 2024-11-23
> **Authors' Response - Part 1**
>
> Thank you for your thorough and thoughtful review of our submission. We greatly appreciate the time and effort you put into providing detailed feedback and suggestions to enhance our work.
>
> > *It should be further shown, either theoretically or experimentally, why the diversity term $\mathcal{L}^{\text{RL}}_{\text{div}} $ in Eq. 5 allows the ensemble Q network to learn the value of diversity.*
>
> Indeed, in the worst case, it would be possible for all members of the ensemble to converge to the same value Q’(s,a) which would differ from r + Q(s,a), and the loss would be satisfied. However, because Q is high dimensional and each member of the ensemble was initialized independently, this case is extremely unlikely and does not happen in practice. Note that this was already true in DENN. Therefore, we do in practice obtain a high diversity among the members of the ensemble. You can verify this directly in Fig. 4: the critic variances for UARL are all >> 1 OOD and are much larger than the critic variances ID, indicating that in practice all the members of the ensemble differ. Note that if they did not differ, the critic variance would be closer to 0, and if UARL was not useful, the critic variance OOD would be much closer to the critic variance ID, as is the case for AWAC (first row of Fig 4). Finally, this result remains consistent as we change the definition of ID and OOD (columns 2 and 3).
>
>
> > *Further clarification is needed for how to calculate $V_Q$ in Algorithm 2 and how to calculate critical variance in uncertainty estimation experiments*
>
> $V_Q$ is calculated simply as the variance of the outputs from the ensemble of critics. For any given state-action pair $(s, a)$, each critic in the ensemble $(Q_1, ..., Q_N)$ will output an estimate of the Q-value, represented as $Q_1(s, a), ..., Q_N(s, a)$. The variance ($V_Q$) is then computed from these Q-value estimates, reflecting the disagreement or uncertainty among the critics for that specific state-action pair. To calculate the variance, we just follow the variance calculation equation:
> $V_Q = \frac{1}{N-1} \sum_{i=1}^{N} (Q_i(s, a) - \bar{Q}(s, a))^2$
> where $\bar{Q}(s, a)$ is the average of the Q-values over the ensemble, $\bar{Q}(s, a) = \frac{1}{N} \sum_{i=1}^{N} Q_i(s, a)$. Given this basic variance formula, the calculation is straightforward and well-established in statistics.
>
> > *The ablation experiments in Appendix B.3 are not detailed enough*
>
> We understand the concern about the readability of the curves and appreciate the suggestion to differentiate the parameter combinations more clearly. Our primary goal with the plots was to convey the effect of different hyperparameter settings on the algorithm's performance without performing extensive hyperparameter tuning. To avoid clutter, we chose to represent all curves with the same color and made the line corresponding to the chosen hyperparameters slightly darker to make it distinguishable. Given the large number of curves (15 in total) with standard deviation as shaded areas, introducing additional colors would likely make the plot overwhelming and harder to interpret.
>
> > *For each $E_i$, the randomized environmental hyperparameter range is determined without a common metric but as a hyperparameter, which may require a lot of time for online tuning for complex scenarios?*
>
> In UARL, the randomization ranges for environmental hyperparameters are not manually tuned or determined via trial and error. Instead, we rely on a dynamic and adaptive approach where the range of each parameter is adjusted based on the agent’s uncertainty in the environment. This process allows the system to automatically adjust the randomization ranges without requiring extensive online tuning. In the paper, we ensure that the parameters are varied in a way that introduces meaningful uncertainty for OOD detection while avoiding excessive complexity or destabilization of the agent. Therefore, while the approach may seem complex, it does not require significant manual intervention or exhaustive online tuning.
>
> > *According to Eq. 5, $R(s, a)$ is not sampled from the dataset $\mathcal{D}$, but the general Q-function update for offline RL, as in Eq. 1, is to use the sampled $r$. Is this a mistake here?*
>
> Thanks for bringing this to our attention. It is indeed a mistake. We fixed it in the revision.

---

> ### Author Response · Authors · 2024-11-23
> **Authors' Response - Part 2**
>
> > *Is there a performance or computational advantage of UARL over direct processing of $E_\omega$'s using the Robust RL algorithm or the algorithm with domain randomization techniques? Can this be illustrated experimentally?*
>
> UARL is designed to handle OOD scenarios without relying on direct interactions with the target environment. Unlike Robust RL or domain randomization, which often require extensive data or access to the full distribution of environments, UARL focuses on modeling uncertainty to generalize effectively across diverse conditions. The goal is to minimize dependence on refining a policy on real-world data, making it safer and more practical for applications where direct interactions with the target environment are risky or infeasible.
>
> > *Notice that in offline training (1st iteration), EDAC performs much worse than CQL and TD3+BC in many environments, which doesn't seem to match the experimental results in the EDAC article?*
>
> We believe this discrepancy may be due to EDAC's potential overfitting to the original D4RL dataset. As mentioned in Section 5, for all experiments and all baselines, we used the implementation and hyperparameters provided at https://github.com/corl-team/CORL, applying the same consistency across other algorithms like EDAC, CQL and TD3-BC.
>
> > *The experiments in this paper were all performed in Mujoco, how do we obtain the real-world demonstration dataset $\mathcal{D}_\omega$ in the simulation environment like Mujoco?*
>
> Since Mujoco is very different from real-world scenarios, there is no real data. In our current experiment, the real-world demonstration dataset $\mathcal{D}_w$ is obtained by running the “expert” policy on the environment with a wide range of values for randomized parameters that differ from the original Mujoco environment, which acts as “real-world data”.
>
> We should have clarified this point further in the paper. Thanks for bringing it into our attention. We included it in the revision, Section 5.

---

> > ### Comment · Reviewer_ruzS · 2024-11-25
> > **Response after rebuttal**
> >
> > Thank you for your thorough and thoughtful response to my comments, as well as for the additional experiments and analysis you have provided. My concerns have been thoroughly addressed, and I have raised the score accordingly. However, I would like to offer a few additional points for your consideration:
> > While I recognize and appreciate your intent to keep the figures simple, I would strongly encourage you to include more detailed ablation studies that rigorously examine the impact of different hyperparameter settings on the performance. Such an analysis would not only enhance the robustness of your findings but also contribute to the overall credibility of your work.
> > With respect to the adaptive range adjustment method you referenced, I kindly request a more comprehensive explanation of its implementation. A more explicit and detailed description would significantly improve the understanding of its mechanics and effectiveness, providing greater clarity for readers.
> > Given the improvements made, I am confident that these additional refinements will further strengthen the quality of the work.

---

> > > ### Author Response · Authors · 2024-11-28
> > > **Authors' Response - After Rebuttal**
> > >
> > > We sincerely thank the reviewer for their fair assessment of our work, and acknowledging our efforts in addressing their concerns and improving the paper.
> > >
> > > > *I would strongly encourage you to include more detailed ablation studies that rigorously examine the impact of different hyperparameter settings on the performance. Such an analysis would not only enhance the robustness of your findings but also contribute to the overall credibility of your work.*
> > >
> > > Thank you for your suggestion. In response, we have added a detailed hyperparameter sensitivity analysis in Appendix B.3, examining the effects of the diversity coefficient and diversity scale across a broad range of values. To enhance clarity, we separated the original plot in the paper by baseline algorithms (AWAC, CQL, and TD3-BC) and hyperparameter $\delta$. This analysis, based on 450 experimental runs, provides valuable insights into the balance between the RL objective and the diversity loss in Equation (5).
> > >
> > > Specifically, the choice of the diversity coefficient $\lambda$ is not arbitrary, as it governs the trade-off between these objectives. When the diversity loss dominates, the agent may prioritize generating diverse behaviors at the expense of achieving the task. Conversely, with smaller $\lambda$ values, the diversity loss becomes negligible, leaving the RL objective largely unaffected. Our findings reveal that moderate $\lambda$ values generally yield the best performance, while extreme values degrade it by skewing this balance.
> > >
> > > These results, visualized with two new figures in Appendix B.3, confirm the robustness of our default hyperparameter configuration and underscore the potential for further optimization through careful fine-tuning in specific scenarios. This enhanced analysis not only validates our findings but also reinforces the credibility and generalizability of our approach.
> > >
> > > > *With respect to the adaptive range adjustment method you referenced, I kindly request a more comprehensive explanation of its implementation. A more explicit and detailed description would significantly improve the understanding of its mechanics and effectiveness, providing greater clarity for readers.*
> > >
> > > Typically, determining the extent of domain randomization is challenging because the true characteristics of the real world are unknown. A common approach is to make educated guesses about these values, train a policy in simulation, and then test it in the real world. If the policy underperforms, one would return to the simulation and increase the degree of randomization.
> > >
> > > In our case, we aim to eliminate the need for real-world validation. An alternative would be to arbitrarily select a randomization range, train within that range, and evaluate the policy using our method. However, this approach of "blind" randomization often results in extensive evaluations until the policy aligns with the target distribution. Instead, we adopted adaptive range adjustment because (1) it provides a structured way to determine the randomization range, and (2) it progressively increases the environment's complexity, allowing the policy to be fine-tuned in subsequent randomized environments without restarting training.

---

### Official Review · Reviewer_FRBA · 2024-10-29

**Soundness:** 2
**Presentation:** 3
**Contribution:** 2
**Rating:** 3
**Confidence:** 5

**Summary:**

This paper deals with the distribution shift issue in reinforcement learning (RL). The authors introduce an approach called Uncertainty-aware Adaptive RL (UARL) that enhances policy generalization across diverse variations of a given environment. UARL views distribution shifts as OOD problems and integrates an OOD detection method to quantify uncertainty, i.e., Q-value variance. UARL realizes diversity in critics via the DENN method. The authors claim that UARL enables iterative policy fine-tuning, starting with offline training on a limited state space and progressively expanding to more diverse variations of the same environment through online interactions. The authors demonstrate the effectiveness of UARL through some experiments on continuous control tasks, showing improved performance and sample efficiency compared to existing methods.

**Strengths:**

## Pros

This paper enjoys the following advantages,

- This paper is well-written. The presentation of this paper is very good and of high quality. The figures are very nice and helpful. Some of the illustration figures significantly convey the idea and core design of UARL, e.g., Figure 1, Figure 2
- This paper is easy to read and easy to follow
- The authors provide open-source codes in the anonymous website, and I believe that the results reported in this paper are reproducible

**Weaknesses:**

## Cons

Despite the aforementioned advantages, this paper has the following flaws.

- **Offline-to-online RL or off-dynamics RL?** This paper claims that they focus on the offline-to-online setting, while it seems that they actually are dealing with off-dynamics RL [1]. The offline-to-online RL typically refers to training a policy offline from a static dataset and then fine-tuning it with some extra online interactions in the same environment. Instead, the authors conduct experiments by modifying the environmental parameters, which essentially constructs dynamics shifts. The experimental setting resembles that in off-dynamics RL [2,3,4]. It seems unclear to me whether it is suitable to name the paper *offline-to-online* or *off-dynamics*.
- **Insufficient related work and Limited novelty.** The authors emphasize that the proposed method can enhance the safety and robustness of RL, but it includes too few related works on safe offline/offline-to-online RL and robust offline/offline-to-online RL. Meanwhile, I have doubts about the novelty of this work. The authors progressively increase hyperparameter randomization of the environment (e.g., friction) when the variance between the Q-ensemble is large and terminates until the policy is safe enough to be deployed. Such an idea resembles [5], which progressively adapts its learned policy by modifying the parameters of the environment. Furthermore, I am a bit confused about the benefits of parameter randomization against domain randomization. If the user can adjust the parameters of the environment, then why not directly use domain randomization? I would expect reasonable justifications for the design of the tasks here. Furthermore, the diversity term is not novel, it is borrowed directly from the existing literature. These all together make the contribution of this paper somewhat limited.
- **Lacking baseline algorithms.** As commented above, this paper claims that it addresses the offline-to-online RL but actually focuses on the off-dynamics RL setting, the authors should include the following baselines,
  - baselines on off-dynamics RL, e.g., [2,3,4]. This is vital to show the effectiveness of the UARL in terms of policy generalization to the target domain
  - RLPD [6], which is a method specially designed for learning with offline data and online interactions with the environment. This baseline is important since it exhibits superior performance given the experimental setting described in this paper (offline data and online interactions). Based on my experience, RLPD can achieve quite strong performance even when there exists dynamics shifts between the offline data and the online environment. Involving this baseline can justify the necessity of the components adopted in UARL (otherwise, one can directly use RLPD for deployment)
  - baselines on safe RL and robust RL. The authors claim that UARL can enhance the safety and robustness of the executed actions, while they do not include any safe RL or robust RL methods for comparison, making it hard to see the rationality and effectiveness of UARL
  - baselines on offline-to-online RL. Unfortunately, this paper also does not include offline-to-online RL methods as valid baseline methods. It is hard to tell the true effectiveness of UARL without these methods, e.g., [7,8,9]
- (minor) **Lacking theoretical justifications.** There is no theoretical analysis of the UARL. I do not want to blame the authors too much on this point. I understand that this paper may set the focus mainly on the empirical side, but including some theoretical analysis can strengthen this paper.
- (minor) **Other issues.**
  - in Equation 5, you wrote $R(s,a)$ in the bellman error, while $r$ in the diversity term $\mathcal{L}_{div}^{RL}$. I think they should be identical, right?
  - the authors do not discuss the limitations of their method in the main text or the appendix. It is important to acknowledge both the advantages and the limitations of the proposed method.
  - the performance improvement of UARL seems limited and incremental on some tasks (e.g., see Figure 3)
  - UARL can still suffer from performance degradation during the fine-tuning phase (e.g., see Figure 5)

Given the above concerns, I vote for rejection since I believe that this paper needs a significant revision before being accepted for possible publication.

[1] Off-dynamics reinforcement learning: Training for transfer with domain classifiers. ICLR

[2] When to trust your simulator: Dynamics-aware hybrid offline-and-online reinforcement learning. NeurIPS

[3] Cross-domain policy adaptation via value-guided data filtering. NeurIPS

[4] Cross-domain policy adaptation by capturing representation mismatch. ICML

[5] Revolver: Continuous evolutionary models for robot-to-robot policy transfer. ICML

[6] Efficient online reinforcement learning with offline data. ICML

[7] Offline-to-online reinforcement learning via balanced replay and pessimistic q-ensemble. CoRL

[8] Bayesian Design Principles for Offline-to-Online Reinforcement Learning. ICML

[9] Proto: Iterative policy regularized offline-to-online reinforcement learning. Arxiv

**Questions:**

Please c.f. the comments above. Besides, I have the following questions,

- In Lines 184-185, you wrote, *disagreement among ensemble models, particularly at the boundary of the training distribution*, what do you exactly mean by *at the boundary of the training distribution*?
- what are the advantages of the diversity term in Equation 5 compared to other diversity terms? (e.g., the diversity term used in the EDAC paper) The authors ought to justify the advantages of doing so rather than using some other methods.
- how can the authors tell that the uncertainty measurement provided in this paper is valid? It would be better to compare against some other uncertainty estimation methods and visualize the uncertainty measurement for a better comparison
- do you have any parameter study on the threshold parameter in Algorithm 2? How it can affect the performance of the agent? Do we need to tune this hyperparameter per task? How can we ensure that the policy is safe when $V_Q \le {\rm threshold}$?

---

> ### Author Response · Authors · 2024-11-23
> **Authors' Response - Part 1**
>
> We sincerely thank the reviewer for their insightful and detailed assessment of our work. We acknowledge that our initial submission did not effectively convey our ideas, and we would like to provide clarification here. Our primary goal is to develop reliable OOD detection capabilities without direct interaction with the target domain. Policy robustness is not the main focus of our work.
>
> > *Offline-to-online RL or off-dynamics RL?*
>
> We thank the reviewer for highlighting this aspect of domain randomization. We acknowledge that it was not clearly explained in our initial submission. Indeed, off-dynamics is a better description of our work.
>
> The prior work [1-5] assumes that the target domain is accessible and relies on refining the policy on the target domain. However, this presents significant safety risks when applied to complex, real-world systems. For example, refining a policy for a self-driving car involves deploying a potentially suboptimal policy to a real system, where the stakes of failure or "termination" are exceptionally high.
>
> In contrast, our work does not rely on refining the policy by interacting with the target domain. Instead, we use an ensemble of critiques as a proxy to evaluate whether the policy is “in-distribution” of the target domain.
>
> > *Furthermore, I am a bit confused about the benefits of parameter randomization against domain randomization*
>
> In conventional domain randomization, there is often no clear guidance on which parameters to randomize or the extent of randomization needed. The process tends to be somewhat arbitrary and relies on iterative validation, comparing the performance of a randomized policy to real-world scenarios. However, this trial-and-error approach also presents significant safety risks when applied to complex, real-world systems, similar to our reasoning above.
>
> Our research seeks to improve domain randomization by introducing a clear evaluation criterion to assess whether a policy is ready for real-world deployment without directly deploying it. We approach this as an out-of-distribution detection problem. We continuously randomize the parameters until the policy is “in-distribution” of the real-world data. By utilizing an ensemble of critics, we evaluate whether the randomization effectively encompasses the true dynamics of the environment. Our concept of safety is based on the fact that we could avoid real-time interactions with the environment to refine the policy.
>
> Our algorithm indeed builds upon the general form of the diversity term introduced in DENN. However, we note that it adapts it to the RL setting by identifying the correct function to repulse from (in our case, $r + \gamma Q(s’, a’)$). Moreover, we extend the original formulation of DENN as we do not require to train in a first phase a “reference function”, making training less cumbersome.
>
> > *Lacking baseline algorithms.*
>
> Thank you for this comprehensive list of suggested baselines. We appreciate the thoroughness of your review and would like to clarify several important distinctions between our work and the suggested comparisons:
> - Fundamental Objective Difference: Our primary goal is to develop reliable OOD detection capabilities, not just policy robustness. While the suggested baselines (H2O, VGDF, PAR) focus on making policies robust to distribution shifts, our work aims to explicitly identify when the system encounters novel situations that require intervention without direct interaction in the novel environment. This is a crucial safety feature for real-world deployment.
> - Real-world Data Usage: Several suggested baselines (H2O, VGDF, Off2On, PROTO) require real-world data during training or policy refinement. Our method deliberately avoids this requirement for safety reasons, using real-world data only for evaluation. This design choice makes our approach more practical for safety-critical applications where real-world training data may be scarce or risky to collect.
> - Complementary Rather Than Competitive: Our method is complementary to many of these baselines rather than directly competitive. While RLPD and other offline-to-online methods focus on policy improvement, our work addresses the crucial preceding question: when is it safe to deploy a policy in the first place? This detection capability could actually enhance the safety of these existing methods.
>
> That said, we acknowledge that adding some robust RL baselines could help demonstrate the indirect benefits of our approach to policy robustness. We will expand our evaluations to include appropriate baselines that do not require real-world training data, focusing on comparing OOD detection capabilities where possible.
>
> > *Lacking theoretical justifications*
>
> While we agree that such analysis could strengthen the paper, our primary focus here is on the empirical evaluation of UARL. We believe the experiments provide strong evidence of its effectiveness, and we will consider theoretical analysis in future work.

---

> ### Author Response · Authors · 2024-11-23
> **Authors' Response - Part 2**
>
> > *in Equation 5, you wrote $R(s,a)$ in the bellman error, while $r$ in the diversity term $\mathcal{L}_{div}^{RL}$. I think they should be identical, right?*
>
> Thanks for bringing this to our attention. It is indeed a mistake. We fixed it in the revision.
>
>
> > *the authors do not discuss the limitations of their method in the main text or the appendix*
>
> We addressed the limitations of UARL in the Conclusion and provided a more detailed discussion in Appendix D of the revised version.
>
> > *the performance improvement of UARL seems limited and incremental on some tasks (e.g., see Figure 3)*
>
> That is a valid observation, as already noted in lines 454 and 455. However, we would like to emphasize again that UARL is not primarily focused on performance improvements. Its main purpose is to assist with OOD detection. The point made in Section 5.1 is to demonstrate that the introduction of UARL does not negatively impact the performance of the underlying algorithms.
>
>
> > *UARL can still suffer from performance degradation during the fine-tuning phase (e.g., see Figure 5)*
>
> This is also a valid observation; however, the balanced replay buffer mechanism in UARL (Section 4.3) results in less performance degradation compared to baseline methods, as shown in Figure 5.
>
> > *In Lines 184-185, you wrote, disagreement among ensemble models, particularly at the boundary of the training distribution, what do you exactly mean by at the boundary of the training distribution?*
>
> By "at the boundary of the training distribution," we refer to data points that lie near the edge of the training data distribution, where the model has less certainty and potentially higher variance in its predictions. These points are typically where the model might struggle to generalize, as they are far from the majority of training data, and thus, where disagreement among ensemble models is most valuable. Introducing diversity at these boundary points (or in OOD regions) leads the ensemble models to be more diverse OOD. We enhanced Section 3.2 in order to make this point clearer.
>
> > *what are the advantages of the diversity term in Equation 5 compared to other diversity terms?*
>
> EDAC aims to prevent over-estimation of Q-values when training with OOD samples, for a conservative training process. This is achieved by diversifying the gradients of Q-values.
>
> UARL enforces the diversity explicitly in the critic output space, which is the space of interest since this is where we compute the critic variance. In this paper, we focused on an empirical evaluation of our approach. Notably, Fig. 4 illustrates the benefits of UARL-enhanced AWAC compared to the base AWAC method. The diversity of UARL leads to a higher ability to discern ID from OOD samples, while maintaining a high performance (Fig. 3).
>
> > *How can the authors tell that the uncertainty measurement provided in this paper is valid?*
>
> Our primary goal with UARL is to detect OOD events rather than to provide an exact estimate of uncertainty. While uncertainty estimation plays a role in guiding the detection process, the focus is on identifying regions where the policy is likely to encounter novel dynamics. We acknowledge that uncertainty calibration is an important consideration for uncertainty estimation methods, but due to the nature of UARL, we do not expect perfect calibration. Instead, we focus on using the uncertainty measurements to highlight areas of high risk and improve OOD detection. We will consider further comparisons with other uncertainty estimation methods in future work, but the current approach is primarily validated through its ability to detect OOD events effectively in our experiments.
>
> > *do you have any parameter study on the threshold parameter in Algorithm 2?*
>
> Threshold at Algorithm 2 acts as a stopping criterion in fine-tuning, allowing the agent to expand the state space until the variance ($V_Q$) of the critic ensemble on the real-world dataset ($D_w$) falls below it. A lower threshold enforces a stricter certainty requirement, enhancing safety by requiring more consistent value estimates but potentially increasing training time. In contrast, a higher threshold could lead to earlier deployment, risking premature exposure to uncertain scenarios. Tuning the threshold may indeed vary per task, as each environment’s dynamics can impact the balance between safety and efficiency. While $V_Q \le \text{threshold}$ provides a proxy for confidence, it is not a formal safety guarantee. True safety would benefit from future work on approaches such as formal verification, explicit safety constraints in learning, or conservative policy updates, which go beyond the current paper’s scope. However, per the reviewer’s suggestion, we will conduct a study on the role of the threshold and its impact on performance.

---

> > ### Comment · Reviewer_FRBA · 2024-11-26
> > **Post-Rebuttal Comments**
> >
> > Sorry for the late response. I thank the authors for providing a rebuttal and revising their manuscript (e.g., including the limitation part). Please find the comments below.
> >
> > > Offline-to-online RL or off-dynamics RL?
> >
> > This paper lies actually in the category of off-dynamics RL. The authors should discuss off-dynamics RL. Prior works like [1,2,3] **do not necessarily assume that the target domain is accessible**. H2O [1] requires an offline target domain dataset and an online source domain environment. VGDF [2] and PAR [3] also conduct experiments when the target domain is fully offline. The authors wrote that these methods can *present significant safety risks when applied to complex, real-world systems*. I disagree with that. [1,2,3] all introduce conservative terms to ensure that the learned policy stays close to the support region of the target domain dataset. These can ensure the safety of the learned policy to some extent. A comparison is needed to see whether UARL can outperform these off-dynamics RL methods. **Please note that I am not requiring extra experiments here.**
> >
> > [1] When to trust your simulator: Dynamics-aware hybrid offline-and-online reinforcement learning. NeurIPS
> >
> > [2] Cross-domain policy adaptation via value-guided data filtering. NeurIPS
> >
> > [3] Cross-domain policy adaptation by capturing representation mismatch. ICML
> >
> > > Related work and novelty
> >
> > I hold my opinion that the related work is insufficient and the novelty of this paper is somewhat weak. The authors should cite more recent offline-to-online RL/robust RL/safe RL/off-dynamics RL papers. The authors wrote that their primary goal is *to develop reliable OOD detection capabilities without direct interaction with the target domain* and policy robustness is not the main focus of their work. I also disagree with this, because the authors emphasize policy safety and robustness numerous times in their paper. I appreciate that the authors include some results comparing UARL against ensemble-based offline RL methods like PBRL and RORL, additional results against safe offline RL ought to be included. I reiterate that the diversity term is not novel, and is borrowed directly from the existing literature. The modifications are minor as listed by the authors.
> >
> > Based on the rebuttal, it seems that the benefit of parameter randomization introduced in this paper against domain randomization is the introduction of a validation criterion. This does not seem to be a clear advantage to me since domain randomization is initially designed to create a variety of simulated environments with randomized properties and train a model that works across all of them, such that the true target environment can be covered. I believe *no clear guidance on which parameters to randomize or the extent of randomization needed* is not an issue or major flaw of domain randomization.
> >
> > > Lacking baselines
> >
> > I hold my opinion that the baseline methods are insufficient. The authors should at least include a comparison against off-dynamics RL methods and RLPD (which I believe is a very important baseline). Comparing against other methods like safe RL algorithms is optional but encouraged. The authors wrote that off-dynamics RL methods are not suitable for comparison because they *focus on making policies robust to distribution shifts*, but I think UARL also does so. The authors also claimed that their method is complementary to baselines rather than competitive. This can be a valid point but is not a good reason to involve too few baseline methods (e.g., not simply comparing X against X+UARL, but X against X+UARL and against X+others and against Y for some algorithms X, Y).
> >
> > > On the performance of UARL
> >
> > I hold my opinion that the performance improvement of UARL seems limited and incremental on some tasks and that UARL can still suffer from performance degradation during the fine-tuning phase (despite the balanced replay buffer mechanism). The limited performance improvement indicates that using UARL does not seem necessary for some tasks, which can be a negative signal for UARL.
> >
> > > What are the advantages of the diversity term in Equation 5 compared to other diversity terms?
> >
> > The authors do not seem to answer my question. I am asking about the advantages of the diversity term in Equation 5 rather than its difference against other methods. Why should we prefer Equation 5 rather than other diversity terms like that used in EDAC?
> >
> > > On the uncertainty measurement
> >
> > If the uncertainty estimate is inaccurate, how can we tell that the OOD detection is reliable? This is extremely vital for UARL to distinguish OOD samples. I hence reiterate that it would be better to compare against some other uncertainty estimation methods and visualize the uncertainty measurement for a better comparison.
> >
> > Overall, I confirm my initial rating and do not favor acceptance at the current stage. It is my hope that the authors can find some of my review and comments helpful in improving the manuscript.

---

> > > ### Author Response · Authors · 2024-11-28
> > > **Authors' Response - After Rebuttal - Part 1**
> > >
> > > We would like to thank the reviewer for getting into a very constructive discussion. We appreciate your feedback and insights to improve our paper.
> > >
> > > > *This paper lies actually in the category of off-dynamics RL.*
> > >
> > > We appreciate the reviewer’s insightful comments and would like to address the point regarding H2O, VGDF, and PAR. While these works do, in fact, assume accessibility to the target domain, we acknowledge their contributions to ensuring safety in off-dynamics RL settings. Below, we provide clarification based on the specific assumptions made by these methods:
> > > - H2O: It fills up half of the replay buffer with target domain data which is used to train the policy, based on the authors' official implementation:
> > >   - https://github.com/t6-thu/H2O/blob/main/SimpleSAC/sim2real_sac_main.py#L185-L186
> > >   - https://github.com/t6-thu/H2O/blob/main/SimpleSAC/sim2real_sac_main.py#L43
> > >
> > >   While it is assumed that the target domain data is accessible offline, the data sensitivity of RL algorithms makes it impractical to train a performant RL agent with only a limited number of data points. However, in UARL, since target domain data is used only for validation, it works well without requiring extensive use of the target domain data during training, making it more efficient in handling limited data availability.
> > > - VGDF: As stated in VGDF’s paper Section 1, the proposed method assumes limited number of online **interactions** with the target domain, while UARL only requires a **limited** pre-collected dataset. For instance, from the VGDF’s Abstract: "we consider the online dynamics adaptation problem, in which case the agent can access sufficient source domain data **while online interactions with the target domain are limited**." Similarly, in Section 1: "In contrast to these works, we consider a more general setting called online dynamics adaptation, where the agent can access sufficient source domain data and **a limited number of online interactions with the target domain**."  Section 3 and Definition 3.1 further clarify this assumption, specifying a 1:10 ratio of online target domain data to source data, amounting to $10^5$ data points **used during the training** (Appendix D.2). In contrast, UARL uses target domain data solely for validation rather than training, achieving strong performance with just 100 trajectories—orders of magnitude fewer than VGDF. This highlights UARL’s efficiency and suitability for data-limited settings.
> > > - PAR: The same criticism directed at VGDF also applies to PAR. As stated in Section 1 of the PAR paper: "We … consider learning policies with sufficient source domain data (either online or offline) and **limited online interactions with the target domain**." Additionally, similar to VGDF, Section 5.1 of the PAR paper specifies a source domain to online target domain data ratio of 1:10, raising similar concerns as those in VGDF.
> > > - Regarding safety risks, while methods like H2O, VGDF, and PAR incorporate conservative regularization terms to keep the learned policy within the target domain’s support region, these methods still pose safety risks in high-stakes, real-world systems. This is because they are not designed to detect when OOD shifts occur; they simply act conservatively within known domains. As highlighted in [1], this distinction is critical—"Robustness" focuses on creating models that are resilient to adversaries, unusual situations, and Black Swan events, while "Monitoring" tracks detecting malicious use, monitoring predictions, and discovering unexpected model functionality. For example, in autonomous driving, while conservative regularization may limit policy deviations, it cannot guarantee safety when the vehicle encounters unforeseen road conditions or new traffic laws. Therefore, while these conservative methods provide some safeguards, they cannot fully mitigate the risks associated with OOD scenarios in complex, high-risk environments.
> > >
> > > Finally, we are uncertain how such a comparison could be conducted without additional experiments, as the reviewer suggests, since H2O, VGDF, and PAR require training policy with target domain data, and our method does not. If the reviewer could provide further clarification or specific suggestions, we would be happy to take the necessary steps to address their concerns.
> > >
> > > [1] Hendrycks, Dan, Nicholas Carlini, John Schulman, and Jacob Steinhardt. "Unsolved problems in ml safety." arXiv preprint arXiv:2109.13916 (2021).

---

> > > > ### Author Response · Authors · 2024-11-28
> > > > **Authors' Response - After Rebuttal - Part 2**
> > > >
> > > > > *Related work and novelty*
> > > >
> > > > Domain randomization lacks clear guidelines on what parameters to randomize and to what extent. For instance, the true value of friction is not directly measurable as it depends on complex dynamic interactions between the robot's materials and the surface it operates on. Similarly, the mass of each robot link, while indirectly measurable, significantly influences overall dynamics. In practice, researchers make assumptions about these parameters, train policies in simulation, and validate them on real systems. However, these assumptions are often flawed–either the randomization range is too narrow or too broad. A limited range can result in unstable policies when deployed on the robot, while an overly broad range may train the policy on scenarios with no real-world relevance, wasting computation and time. This iterative trial-and-error process can take days or even weeks to identify an effective randomization strategy.
> > > >
> > > > For this reason, our goal is to develop criteria to skip the online validation step, so we can make sure we do not execute a potentially problematic policy on a real system.
> > > >
> > > > Regarding the novelty of the diversity term, we acknowledge that it draws inspiration from existing methods in the supervised learning literature. However, our contribution lies in adapting this concept to the RL setting, where enforcing diversity in the output space, rather than the parameter space, presents unique challenges and implications. By validating that the insights from the original DENN paper are both applicable and effective in the UARL context, we demonstrate the value and utility of this adaptation. This extension is non-trivial, as it addresses the specific dynamics of RL environments and shows that these insights can significantly enhance OOD detection and policy robustness in our framework. While we acknowledge and appreciate the reviewer's perspective, we respectfully disagree for the reasons outlined above.
> > > >
> > > > > *Lacking baselines*
> > > >
> > > > As highlighted in our earlier responses and addressing the reviewer’s specific concern about off-dynamics RL, our method is fundamentally distinct from both off-dynamics approaches and RLPD. Specifically focusing on RLPD, this method is primarily centered around **online learning** while leveraging offline data to enhance the process. In contrast, the core principle of UARL lies in its complete avoidance of any learning that utilizes data from the target domain. This distinction is crucial and underscores why a direct comparison is unfair.
> > > >
> > > > To clarify further, please refer to Algorithm 1, lines 12–14 of the RLPD paper, where half of the replay buffer is explicitly filled with samples from online interactions with the target domain. By comparison, our approach assumes access only to a limited set of datapoints from the target domain, without any direct online interactions. In this regard, RLPD aligns more closely with off-dynamics RL methods such as VGDF, PAR, and H2O, as discussed earlier. By eliminating the need for learning with target domain data, UARL avoids these challenges when deploying an RL policy to the real-world.
> > > >
> > > > > *On the performance of UARL*
> > > >
> > > > Thank you for your comment. As explained in our earlier comments, the main focus of our work is improving OOD detection for RL agents, not enhancing their performance. We stated this point clearly in the revision (line 75), our key contribution is “a method for quantifying uncertainty and adapting policy without direct interactions in the OOD environments.” Performance improvement is not the primary objective of this work. Instead, we present performance results to demonstrate that our method does not compromise the underlying policy’s effectiveness and, in fact, achieves competitive performance with the addition of diversity term. Our primary goal remains OOD detection, as demonstrated in Figure 4, which highlights the clear separation between ID and OOD scenarios achieved by UARL.

---

> > > > > ### Author Response · Authors · 2024-11-28
> > > > > **Authors' Response - After Rebuttal - Part 3**
> > > > >
> > > > > > *What are the advantages of the diversity term in Equation 5 compared to other diversity terms?*
> > > > >
> > > > > The diversity term in Equation 5 of UARL is designed specifically to enhance the **critic variance** in the Q-value output space, which is crucial for the OOD detection framework of our method. The critic output space is where we detect OOD events, so this learning rule is directly aligned with the intended objective of OOD detection. Unlike the diversity term in EDAC, which focuses on reducing gradient similarity across ensemble networks to improve Q-value accuracy and prevent overestimation, the UARL diversity term aims to **maximize variability in Q-values** to better separate ID and OOD data.
> > > > >
> > > > > This distinction is critical: EDAC's diversity term serves a **primarily conservative Q-learning goal**, penalizing OOD actions by leveraging variance, thereby avoiding overestimation errors. In contrast, UARL’s diversity term is not concerned with conservative Q-value estimation but is explicitly crafted to amplify differences in Q-values to improve **uncertainty quantification and OOD detection**. This tailored focus makes the UARL diversity term uniquely suitable for our framework, as it aligns with our primary objective of robust OOD detection rather than mitigating overestimation bias.
> > > > >
> > > > > > *On the uncertainty measurement*
> > > > >
> > > > > We appreciate the reviewer’s concern regarding the importance of reliable uncertainty estimation for effective OOD detection. As included in the revision, we have already included comparisons with OOD-aware baselines, such as PBRL and RORL, in Appendix B.5. To further strengthen our argument, we have now added additional comparisons with EDAC and DARL [1], which are specifically focused on uncertainty estimation in Appendix B.5. These comparisons provide a more comprehensive evaluation of the effectiveness of our method in distinguishing OOD samples.
> > > > >
> > > > > [1] Zhang, Hongchang, Jianzhun Shao, Shuncheng He, Yuhang Jiang, and Xiangyang Ji. "DARL: distance-aware uncertainty estimation for offline reinforcement learning." AAAI 2023.

---

### Official Review · Reviewer_1mvx · 2024-10-31

**Soundness:** 3
**Presentation:** 2
**Contribution:** 3
**Rating:** 5
**Confidence:** 3

**Summary:**

The paper introduces Uncertainty-aware Adaptive RL (UARL), an innovative framework to tackle distributional shifts and out-of-distribution (OOD) issues when deploying reinforcement learning (RL) policies in real-world environments. This is accomplished by implementing OOD detection to quantify policy uncertainty and iteratively refine high-uncertainty regions (of the state space), adapting the policy
for safe and effective performance deployment. UARL demonstrates several notable advancements,
- A method for quantifying policy uncertainty using OOD detection.
- An offline-to-online (O2O) adaptation strategy that balances online and offline data, utilizing a diverse ensemble of critics to better handle distributional shifts.
- Experiments on MuJoCo continuous control tasks that validate UARL’s effectiveness in terms of performance, robustness, and sample efficiency.

**Strengths:**

- UARL presents a compelling approach to address the challenges of deployment a policy in RL. The progressive expansion of state space via repulsive locations and a balanced replay buffer to manage data distribution shifts are novel and theoretically sound.
-The usage of an ensemble of diverse critics to perform OOD detection and policy refinement represents a robust methodology that has support from the experimental result

**Weaknesses:**

- The paper could better highlight its unique contributions compared to existing OOD and ensemble-based offline RL methods. A clearer differentiation of UARL's specific advancements would help underscore its novelty within the landscape of similar approaches.

- The experimental validation, limited to few environments such as the Ant-v4 and HalfCheetah-v4 environments, may not fully capture the method’s effectiveness across a diverse range of tasks. Extending the experiments to include more varied environments would provide a more comprehensive assessment and enhance the generalizability of the results.

- A comparison with recent state-of-the-art methods, such as PBRL[1], RORL[2], would strengthen the empirical evaluation. By benchmarking UARL against PBRL and similar approaches, the paper could provide a more robust validation of its improvements in uncertainty handling and performance stability.

[1] Pessimistic Bootstrapping for Uncertainty-Driven Offline Reinforcement Learning
[2] RORL: Robust Offline Reinforcement Learning via Conservative Smoothing

**Questions:**

1. Comparing the computational overhead of your method with that of baseline algorithms would strengthen your work. Could you include this information to provide a clearer understanding of its efficiency?

1. Does this algorithm fall within the scope of Offline Reinforcement Learning? If so, it would be helpful to clarify its placement within the Offline Reinforcement Learning landscape. Enhancing the abstract and introduction to better position the algorithm within this broader context would significantly improve the clarity and impact of your paper.

I am open to raising my score based on these improvements.

---

> ### Author Response · Authors · 2024-11-23
> **Authors' Response - Part 1**
>
> We sincerely thank you for your insightful feedback and detailed comments on our work. Your suggestions have significantly contributed to improving the clarity and presentation of our research.
>
> > *The paper could better highlight its unique contributions compared to existing OOD and ensemble-based offline RL methods. A clearer differentiation of UARL's specific advancements would help underscore its novelty within the landscape of similar approaches.*
>
> We understand the reviewer’s concerns regarding the framing and narrative of the paper in its current format. We will work to adjust the structure and focus to clarify our main contributions and streamline the discussion around key topics. Specifically, we will refine the emphasis on our central problem domain to create a more cohesive and targeted narrative, reducing the scope of secondary discussions where possible. The changes are reflected in Sections 1 and 2 in the revision.
>
> > *The experimental validation, limited to few environments such as the Ant-v4 and HalfCheetah-v4 environments, may not fully capture the method’s effectiveness across a diverse range of tasks. Extending the experiments to include more varied environments would provide a more comprehensive assessment and enhance the generalizability of the results.*
>
> While we believe our experiments provide a solid foundation for validating our method's core principles, we acknowledge the value of broader testing. That is why we are working on providing results on more environments to improve the generalizability of our findings.
>
> > *A comparison with recent state-of-the-art methods, such as PBRL[1], RORL[2], would strengthen the empirical evaluation.*
>
> We want to clarify an important distinction between our work and RORL/PBRL that we should have emphasized more clearly in the paper. While RORL and PBRL represent significant advances in robust offline RL, they tackle a fundamentally different problem than our work. These methods focus on building robust policies that can maintain performance despite encountering OOD scenarios, primarily through uncertainty-based penalization during training. This is distinct from our core objective: developing reliable mechanisms to **detect** when a system is operating in OOD conditions.
>
> The distinction becomes clear when considering real-world applications: a robust policy might continue operating in OOD conditions (as RORL/PBRL aim to achieve), but this could be undesirable in safety-critical systems where we need to explicitly recognize such situations and potentially halt operation or seek human guidance. Our method's progressive environmental randomization serves to build this detection capability, training the uncertainty estimator to recognize the boundaries between familiar and novel situations.
>
> While our approach does yield some robustness benefits through its iterative fine-tuning, this is secondary to its main purpose of reliable OOD detection. We revised the paper to better articulate this fundamental difference in objectives and clarify how our method specifically targets the detection challenge rather than just robustness, reflected in Appendix C.
>
> We deemed PBRL and RORL are not directly comparable to our work; nevertheless, we included results for a limited experimental setting comparing our method with PBRL and RORL in Appendix B.5 of the revised paper for the readers’ reference.
>
> > *Comparing the computational overhead of your method with that of baseline algorithms would strengthen your work. Could you include this information to provide a clearer understanding of its efficiency?*
>
> Our method does introduce additional computational complexity, primarily through the repulsive dataset, which impacts memory usage more significantly than computation time. Most baseline methods (CQL, AWAC, TD3+BC) already use dual-critic architectures, so our additional computational overhead stems from maintaining separate nominal and repulsive datasets, the diversity loss calculation, and the replay buffer balancing mechanism.
>
> Our measurements suggest the memory usage was increased by approximately 50% compared to baseline methods, primarily due to storing both nominal and repulsive datasets. That is also subject to change given the desired ratio of nominal data points to repulsive ones. The computational time overhead is relatively modest, in the 10-20% range, and is equivalent to training a regular actor-critic method with a larger batch size
>
> Our results suggest that uncertainty awareness provides significant value that outweighs the additional computational requirements.
> We included these results in Appendix D of the revision.

---

> ### Author Response · Authors · 2024-11-23
> **Authors' Response - Part 2**
>
> > *Does this algorithm fall within the scope of Offline Reinforcement Learning?*
>
> We believe this topic is already thoroughly addressed in the Related Work section (Lines 110-114), where we explicitly discuss the distinctions between our approach and existing Offline RL methods. However, we recognize the importance of ensuring that this contextualization is evident throughout the paper. To address this, we further emphasized these points in the abstract.

---

> > ### Author Response · Authors · 2024-11-28
> > **Authors' Comment**
> >
> > As the discussion period draws to a close, we wanted to follow up to check if you have had a chance to review our rebuttal. If you have any remaining questions or concerns, we would be glad to address them. Thank you.

---

> ### Comment · Reviewer_1mvx · 2024-11-28
> **Feedback to the authors**
>
> Thank you for your feedback. After revisiting the reviews, I believe my score aligns with the feedback and will keep it. I appreciate your effort and encourage you to continue developing your ideas.

---

### Official Review · Reviewer_e1VH · 2024-11-03

**Soundness:** 3
**Presentation:** 3
**Contribution:** 2
**Rating:** 3
**Confidence:** 3

**Summary:**

In this paper, a novel RL pipeline, Uncertainty-aware Adaptive RL (UARL), has been proposed to enhance policy generalization across diverse variations of a given environment. UARL frames distribution shifts as OOD issues and integrates a new OOD detection method to quantify uncertainty. This method enables iterative policy fine-tuning, beginning with offline training on a limited state space and gradually expanding to more diverse variations of the same environment through online interactions.

**Strengths:**

The problem raised in this paper is important and the experiments are solid.

**Weaknesses:**

Weakness:

1) The core problem in this paper, i.e., distributional shift, and many important concepts are not discussed in detail. In the introduction, the authors discuss about the theoretical shortcomings of robust RL, safe RL, and the distributional shift in offline2online RL, but they ignore the discussion about the relationships between these concepts. For example, what is the relationship between the robust RL and the distributional shift in offline setting? Furthermore, what is the difference between the distributional shift problems in offline RL and offline2online RL settings? Why the proposed method could successfully solve the problem of distributional shift? Please answer this question from a high-level view.

2) In offline (to online) RL, the uncertainty quantifier is defined clearly as the upper bound of the error produced by the empirical Bellman operator (see [1], Eq.(4.1)). Then we may concern that whether the uncertainty defined in this paper's Eq.(5) has the relationship with the uncertainty quantifier as we have known in [1]? Does it a valid uncertainty quantifier theoretically? The author should discuss about this point.

[1] Jin. et al., Is Pessimism Provably Efficient for Offline RL.

3) In offline RL, there have been many uncertain-aware methods to deal with the distributional shift problem, such as [2] and [3]. In the list two works, they both penalize the OOD actions by the constructed uncertainty quantifiers. So in our view, the method in this work is not beyond the scope of these methods and lack of the sufficient discussion with the advantage over the existing uncertain-aware methods.

[2] Bai. et al., Pessimistic bootstrapping for uncertainty-driven offline reinforcement learning.
[3] Sun. et al., Model-Bellman Inconsistency for Model-based Offline Reinforcement Learning.

4) the convergence property of the proposed algorithm should be discussed, especially line 8 in Algorithm 2 - what if this condition is never voilated?

**Questions:**

see above.

---

> ### Author Response · Authors · 2024-11-23
> **Authors' Response - Part 1**
>
> Thank you for your detailed review and for engaging with our work. We appreciate the time and effort you put into your thoughtful feedback. We realize there may have been some miscommunication about the problem we are addressing, and we have taken steps to clarify our goals and contributions in the revised manuscript. Your comments have helped us identify areas where clearer explanations were needed.
>
> > *The core problem in this paper, i.e., distributional shift, and many important concepts are not discussed in detail.*
>
> We acknowledge that the relationships between distributional shift across different RL paradigms deserve a more thorough treatment. Our work primarily addresses detecting distributional shift during offline-to-online transition, which differs from purely offline RL in a crucial way: offline RL faces fixed distributional gaps between training and deployment data, while offline-to-online transition must handle dynamic shifts as the policy begins interacting with the environment. Our method addresses distributional shift through progressive environmental randomization that systematically expands the policy's exposure to different dynamics. Unlike robust RL which typically optimizes for worst-case performance across a fixed distribution, we actively shape the distribution of experiences to build reliable uncertainty estimates. This helps the policy identify when it's encountering novel scenarios and adapt appropriately. We will revise the introduction to clarify these conceptual relationships and better motivate how our approach bridges the gap between offline training and online adaptation.
>
> > *In offline (to online) RL, the uncertainty quantifier is defined clearly as the upper bound of the error produced by the empirical Bellman operator (see [1], Eq.(4.1)).*
>
> In both papers, the goal of quantifying uncertainty is to identify states and actions where the learned policy might not be reliable due to limited data or distributional shifts. While Jin et al.’s uncertainty quantifier provides a theoretical upper bound on Bellman operator errors, our variance-based metric (Equation 5) offers a practical and adaptive approach for estimating uncertainty, particularly effective in offline-to-online RL. Both methods aim to identify unreliable state-action regions, but our ensemble variance highlights critic disagreement, serving as a scalable proxy for uncertainty under distributional shifts. Though not grounded in formal bounds like Jin et al., our method has demonstrated robust empirical performance. Future work could further investigate its theoretical properties, potentially bridging the gap between heuristic and formal uncertainty quantification.
>
> > *In offline RL, there have been many uncertain-aware methods to deal with the distributional shift problem, such as [2] and [3].*
>
> While MOBILE and PBRL make valuable contributions to robust offline RL, there is a fundamental difference in objectives and methodology. MOBILE and PBRL focus on robustness against OOD scenarios by penalizing uncertain actions during offline training. Their primary goal is to learn conservative policies that avoid OOD situations. This is achieved through uncertainty-based penalties in Q-value estimation and specialized sampling techniques.
> In contrast, our work addresses a distinctly different challenge: explicit OOD detection during deployment, **without direct interactions with the OOD scenarios**. Rather than just avoiding OOD actions, we aim to actively identify when the agent encounters novel scenarios. This capability is crucial for:
>   1. Maintaining awareness of when the current policy might be unreliable
>   2. Enabling informed decisions about when to request human intervention
>   3. Guiding targeted data collection for policy improvement
>
> Our progressive environmental randomization approach specifically builds up the agent's ability to distinguish between in-distribution and OOD states, rather than just being robust to them. While robustness emerges as a beneficial side effect of our method during iterative fine-tuning, it is not the primary objective.
> We provided a discussion in Appendix C of the revision to more clearly articulate this distinction between robustness-focused approaches (like MOBILE and PBRL) and our detection-focused methodology.

---

> ### Author Response · Authors · 2024-11-23
> **Authors' Response - Part 2**
>
> > *the convergence property of the proposed algorithm should be discussed, especially line 8 in Algorithm 2 - what if this condition is never voilated?*
>
> If the condition in line 8 of Algorithm 2 is never violated, it likely means that the domain randomization is ineffective. High uncertainty ($V_Q$) despite diverse training environments suggests that the agent has not learned a policy that generalizes to real-world conditions. This indicates that either the domain randomization strategies are inadequate or the real-world data ($D_w$) is too different from the simulated environments. In such cases, continuing training would not be productive. Instead, it would require revisiting the domain randomization process and real-world data to ensure they are sufficiently representative of the target environment. It is important to note that this is not a shortcoming of our method, but rather a reflection of the limitations of the domain randomization process or the quality of the real-world data. Our approach assumes that these factors are appropriately addressed, and failure in this regard would require improvements outside the scope of our method. We highlighted these points in the revision at the end of Section 4.4.

---

> ### Author Response · Authors · 2024-11-28
> **Authors' Comment**
>
> As the discussion period draws to a close, we wanted to follow up to check if you have had a chance to review our rebuttal. If you have any remaining questions or concerns, we would be glad to address them. Thank you.

---

### Official Review · Reviewer_aVXx · 2024-11-03

**Soundness:** 3
**Presentation:** 3
**Contribution:** 3
**Rating:** 6
**Confidence:** 4

**Summary:**

The paper proposes a novel pipeline to detect OOD environment variations and gradually fine-tunning the agent until high confidence safe deployment is possible.

**Strengths:**

1. The overall method is well-motivated and clearly stated.
2. Weighting samples in the normal dataset and repulsive dataset differently is intuitive and is demonstrated to be effective empirically.
3. Using a set of critiques and its variances as a measure of environmental uncertainty explore new possibilities from the existing DENN method.

**Weaknesses:**

1. Repulsive locations discussion could be written more formally and thus more concisely. The current version is a bit too dense verbally. I'm also a bit confused by Figure 2, while it's a nice visual, is it something derived from the experiments or is it just a conceptual illustration?
2. Lack of related works: changing environment parameters to achieve repulsive locations is quite related to the literature on curriculum learning. A good survey to start with is https://arxiv.org/pdf/2003.04960. Also, blindly varying the environmental parameters may lead to unexpected harmful environments dampening the agent training: https://openreview.net/forum?id=hp4yOjhwTs&noteId=vZMeHQbnJK
I would suggest authors add a subsection for curriculum reinforcement learning in the related work for a more thorough introduction to the problem backgrounds.

**Questions:**

1. Typo: Algo 1 line 7, should it be "OOD" instead of "ODD"?
2. How do you define "progressively expanding the randomization range" for different environment parameters? More specifically, increasing friction by 1% and increasing the agent's mass by 1% may have vastly different impacts on the task difficulties. Could you discuss more on the relative impact of changing each parameter to the environment difficulty?

---

> ### Author Response · Authors · 2024-11-23
> **Authors' Response**
>
> We sincerely thank you for your assessment of our work. Your comments have helped us further strengthen the clarity and presentation of our contributions. Following are our responses to your comments.
>
>
> > *Repulsive locations discussion could be written more formally and thus more concisely. The current version is a bit too dense verbally.*
>
> The section aims to establish a clear conceptual framework for how our approach progressively expands the exploration space through targeted environmental randomization. While we appreciate the suggestion for potential rephrasing, we believe the current format is necessary as we introduce the concept of repulsive locations in the RL context, detail our specific implementation through hyperparameter randomization, illustrate the progressive expansion of the exploration space through Fig. 2, and distinguishes our targeted approach from standard domain randomization. However, we welcome specific suggestions for improving the formality or conciseness of particular passages while maintaining these core explanatory elements. If the reviewer has specific areas where you feel the language could be tightened, we would be happy to address them in revision.
>
> > *I'm also a bit confused by Figure 2, while it's a nice visual, is it something derived from the experiments or is it just a conceptual illustration?*
>
> It is a conceptual visualization for illustrating the idea. It did not derive from actual experiments but is meant to provide an overview of the approach. We mentioned that explicitly in the revision.
>
> > *Lack of related works*
>
> UARL differs fundamentally from curriculum RL, which relies on a structured progression of tasks, helping the agent learn incrementally by tackling simpler tasks before harder ones. In curriculum RL, the policy is refined on new tasks as they are introduced. However, this sequential approach may fall short in real-world scenarios where unexpected, OOD events can disrupt performance. In contrast, UARL dynamically detects and adapts to OOD shifts based on real-time uncertainty rather than a fixed curriculum. Notably, UARL assumes that policies cannot be refined in the target environment. By quantifying and addressing uncertainty, our approach improves safety and robustness in unpredictable environments, making it better suited for real-world deployment where adaptability is key. To highlight these differences further, we will provide a discussion in the Related Work section.
>
> > *How do you define "progressively expanding the randomization range" for different environment parameters?*
>
> In UARL, "progressively expanding the randomization range" does not involve a simple, uniform increase in each parameter. Instead, we dynamically adjust the randomization based on environmental uncertainty, without requiring expert knowledge for each parameter. Specifically, the expanded dataset serves as a repulsive dataset, introducing uncertainty to help detect OOD cases. During the verification process on the real-world environment ($\mathcal{D}_w$), two outcomes guide us: if low uncertainty is detected, it suggests that the dataset adequately captures the real-world dynamics, meaning the parameter range is reasonable. If high variance remains, it indicates that the dataset is still "too narrow," signaling that the randomization range should be further expanded to better encompass real-world variability. This iterative process ensures that we avoid training in environments that are too far from the actual dynamics, stopping before the agent encounters destabilizing conditions.

---

### Author Response · Authors · 2024-11-23
**Message to all reviewers**

We want to thank all five reviewers for taking the time to provide thoughtful and detailed feedback on our paper. Your comments have been incredibly helpful in improving the quality of our work, and we truly appreciate your effort and expertise.
The revision will be uploaded in 24 hours. In the revised version of the paper, we have worked to address many of the concerns and suggestions you raised, with changes written in red. We hope you will find these updates helpful as you review the changes.

---

### Meta-Review · Area_Chair_J8qH · 2024-12-22

**Metareview:**

The paper prpoposes Uncertainty-aware Adaptive RL (UARL) to enhance policy generalization across diverse variations of a given environment. UARL frames distribution shifts as OOD generalization and uses an OOD detection method to quantify uncertainty. The main weaknesses of the paper pertain to lack of simplicity of the approach, comparison to other works in off-dynamics RL, lack of ablations of all components of the approach, and unclear effectiveness of their approach as tasks are scaled further. More concretely, personally I think that the claim "UARL can generalize to OOD environments without running policies in the OOD environment" needs more justification. It is also unclear how one would tune hyperparameters of the method in an actual real-world deployment, which is the motivation of this approach.

Given the competitiveness of papers this year, and so many unclear bits surrounding this paper, I agree with the reviewers' overall opinions and decide to unfortunately reject the paper.

**Additional Comments On Reviewer Discussion:**

Reviewer FRBA raised some interesting points, largely some which I agree with. I would suggest that authors look at these points and try to incorporate major changes to ensure that comparisons to other baselines and methods are accounted for. While I agree that there are minor differences between algorithms / settings, please also do note that these comparisons will also generally help readers and practitioners make better sense of your method.

---

### Decision · Program_Chairs · 2025-01-22

Reject